# The short isoform of the host antiviral protein ZAP acts as an inhibitor of SARS-CoV-2 programmed ribosomal frameshifting

Matthias M. Zimmer[1,4], Anuja Kibe [1,4], Ulfert Rand [2], Lukas Pekarek[1], Liqing Ye[1], Stefan Buck[1], Redmond P. Smyth [1,3], Luka Cicin-Sain [2] & Neva Caliskan [1,3✉]

Programmed ribosomal frameshifting (PRF) is a fundamental gene expression event in many viruses, including SARS-CoV-2. It allows production of essential viral, structural and replicative enzymes that are encoded in an alternative reading frame. Despite the importance of PRF for the viral life cycle, it is still largely unknown how and to what extent cellular factors alter mechanical properties of frameshift elements and thereby impact virulence. This prompted us to comprehensively dissect the interplay between the SARS-CoV-2 frameshift element and the host proteome. We reveal that the short isoform of the zinc-finger antiviral protein (ZAP-S) is a direct regulator of PRF in SARS-CoV-2 infected cells. ZAP-S over-expression strongly impairs frameshifting and inhibits viral replication. Using in vitro ensemble and single-molecule techniques, we further demonstrate that ZAP-S directly interacts with the SARS-CoV-2 RNA and interferes with the folding of the frameshift RNA element. Together, these data identify ZAP-S as a host-encoded inhibitor of SARS-CoV-2 frameshifting and expand our understanding of RNA-based gene regulation.

[1] Helmholtz Institute for RNA-based Infection Research (HIRI), Helmholtz Zentrum für Infektionsforschung (Helmholtz Centre for Infection Research), Josef-Schneider-Strasse 2, 97080 Würzburg, Germany. [2] Helmholtz Zentrum für Infektionsforschung, Inhoffenstrasse 7, 38124 Braunschweig, Germany. [3] Medical Faculty, Julius-Maximilians University Würzburg, 97074 Würzburg, Germany. [4] These authors contributed equally: Matthias M. Zimmer, Anuja Kibe. ✉email: neva.caliskan@helmholtz-hiri.de

The novel severe acute respiratory syndrome-related coronavirus (SARS-CoV-2), the causal agent of coronavirus disease 2019 (COVID-19), emerged rapidly to become a global threat to human health[1]. Analysis of RNA- and protein-interaction networks have rapidly improved our understanding of SARS-CoV-2 viral replication[2,3]. However, detailed mechanistic understanding of the interplay between RNA-protein complexes, which could inform the design of novel antivirals, is still lacking. Here, functionally important RNA elements of the SARS-CoV-2 genome represent ideal targets due to their evolutionary conservation. One of those well-conserved RNA elements is the programmed ribosomal frameshift site.

A hallmark of SARS-CoV-2 and many other viruses is the −1 programmed ribosomal frameshifting (−1PRF) event which allows translation of multiple proteins from the same transcript. Frameshifting increases the coding potential of genomes and is often used to expand the variability of proteomes or to ensure a defined stoichiometry of protein products[4,5]. In coronaviruses, −1PRF on the *1a/1b* gene is essential for efficient viral replication and transcription of the viral genome. In cells, the efficiency of this frameshifting event varies between 20 and 40%[6,7]. PRF relies on the presence of a slippery heptameric sequence (in coronaviruses U UUA AAC) and a stimulatory RNA secondary structure, such as a pseudoknot (PK) (Fig. 1A). Mutations in the slippery sequence and downstream RNA structure drastically impair frameshifting efficiency (FE)[8,9].

Traditionally, efforts to understand the mechanism of −1PRF focused on *cis*-acting modulatory elements. Previous work in in vitro reconstituted translation systems revealed in detail how ribosome pausing on the slippery codons may lead to a kinetic partitioning between the two reading frames and favor movement of translating ribosomes to the alternative reading frame[6,10]. It has been shown that −1PRF canonically occurs during a late stage of the tRNA translocation step. The stimulatory RNA element causes ribosomes to become trapped in an unusual conformation that is relieved by either the spontaneous unfolding of the blockade or by a −1 slip on the mRNA[6,10]. It is now becoming clear, however, that *cis*-acting elements are not the only determinants of frameshifting in cells and that *trans*-acting viral and cellular factors as well as small molecules or oligonucleotides can alter frameshifting levels[11–14]. Despite these recent insights, fundamental questions remain unanswered. For example, it is still unclear how important RNA-binding factors are for frameshifting processes in general and how exactly interactions of these factors with the RNA alter the mechanical properties of RNA or the choice of the reading frame.

At least three plausible routes to modulate frameshifting by *trans*-acting factors can be envisioned. First, the binding of the factor can transform the downstream RNA element to a more stable roadblock. This has been shown for cardiovirus 2 A, poly-(C) binding protein and some small molecules such as naphthyridine carbamate tetramer (NCTn)[11,12,15]. In these cases, the specific interaction of the factor with the nucleotides downstream of the slippery codons leads to an increase in frameshifting. Second, the factor can target stalled ribosomes, as was shown for eukaryotic release factors such as eRF1 alone or eRF1/3, which are recruited by Shiftless (SHFL) to the HIV-1 frameshift site[16,17]. Different from the first group of regulators, the interaction of both SHFL and release factors is not dependent on the identity of the frameshift RNA. Therefore, it remains to be solved how the frameshifting ribosome complexes are recognized by these *trans*-acting factors. A possible third route might act via remodeling or destabilization of the frameshift RNA elements through direct interactions with the *trans*-factor. So far, however, no cellular or viral factor has been reported to affect FE through that route.

Given the importance of −1PRF for the SARS-CoV-2 life cycle, we set out to comprehensively identify direct protein interactors of the SARS-CoV-2 frameshift RNA element using an in vitro RNA-antisense capture and mass spectrometry-based screen[18]. Through this approach, we identified the short isoform of the zinc-finger antiviral protein (ZAP-S, ZC3HAV1) as the most prominent interaction partner amongst over 100 proteins detected. We demonstrated that ZAP-S acts as a host-encoded inhibitor of SARS-CoV-2 *1a/1b* frameshifting in vivo and in vitro. Intriguingly, ZAP-S overexpression reduced the replication of SARS-CoV-2 by about 20-fold. Apart from SARS-CoV-1 and 2, we were not able to identify other PRF sites that are affected by ZAP-S, which suggests that certain RNA elements are preferentially recognized by ZAP-S. Using a multidisciplinary approach, we further revealed important molecular principles of frameshifting downregulation by ZAP-S. Specifically, we show that ZAP-S can alter the physical properties of the frameshift RNA. Our study highlights that the expression of the SARS-CoV-2 ORF1a/1b can be directly modulated by a host-encoded RNA-binding protein (RBP) during infection. These findings provide new insights on −1PRF regulation and the interplay between SARS-CoV-2 replication and host defense, thereby paving the way for novel RNA-based therapeutic intervention strategies.

## Results

**SARS-CoV-2 PRF RNA capture identifies novel host interactors**. To identify potential cellular RBPs that interact with the −1PRF element of SARS-CoV-2, an in vitro synthesized RNA fragment corresponding to nucleotides 13456-13570 of the SARS-CoV-2 genome was incubated with lysates of SARS-CoV-2 infected and uninfected Calu-3 cells and naïve HEK293 cells (Fig. 1A, B)[18]. Calu-3 cells are lung epithelial cells that are commonly used to study CoV infection[19]. HEK293 cells are routinely used to study RNA-protein interactomes and represent an ideal system to assess possible cell-type specific interactions[20]. To exclude any non-specific binders, an 80 nucleotides long non-structured RNA was employed as a control. RNAs were captured by a biotinylated antisense DNA-oligo, and interacting proteins were identified by LC-MS/MS (liquid chromatography tandem mass spectrometry) analysis (Fig. 1B, C).

We identified more than 100 proteins that were at least twofold enriched compared to the control RNA. According to GO term analysis, the majority (80%) of our hits are categorized as RBPs (Supplementary Fig. 1A). For example, we observed enrichment of the viral nucleocapsid protein (N) in infected lysates, which is a well-described RBP[21]. 35% and 30% of the enriched RBPs were involved in splicing and ribosome biogenesis, respectively (Supplementary Fig. 1A). Among those, 9 proteins were identified in infected and uninfected Calu-3 cells as well as in HEK293 cells, 19 proteins were common to infected and uninfected Calu-3 cells, 18 hits were found only in HEK293 cells, 15 were captured only in uninfected Calu-3 cells, and 40 were present only in infected Calu-3 cells (Supplementary Fig. 1B). The core interactome of 9 proteins identified in all three cell systems encompasses well-described post-transcriptional regulators, namely HNRNPH1, DHX36, GRSF1, HNRNPH2, HNRNPF, ZAP, MATR3, ELAVL1 and POP1 (Fig. 1C, D, Supplementary Fig. 1C). Based on their enrichment and functions in translational regulation, we selected 20 proteins for downstream analysis and functional characterization. These candidates were also hits recently identified in SARS-CoV-2 genome-wide RNA interactome studies (Supplementary Fig. 1D)[3,21–23]. Several of those interactors have been shown to play a role in RNA processing, including splicing (such as HNRNPs F, H1, and H2), RNA trimming (POP1) and RNA surveillance (ZAP)[24–26]. Translational regulators included

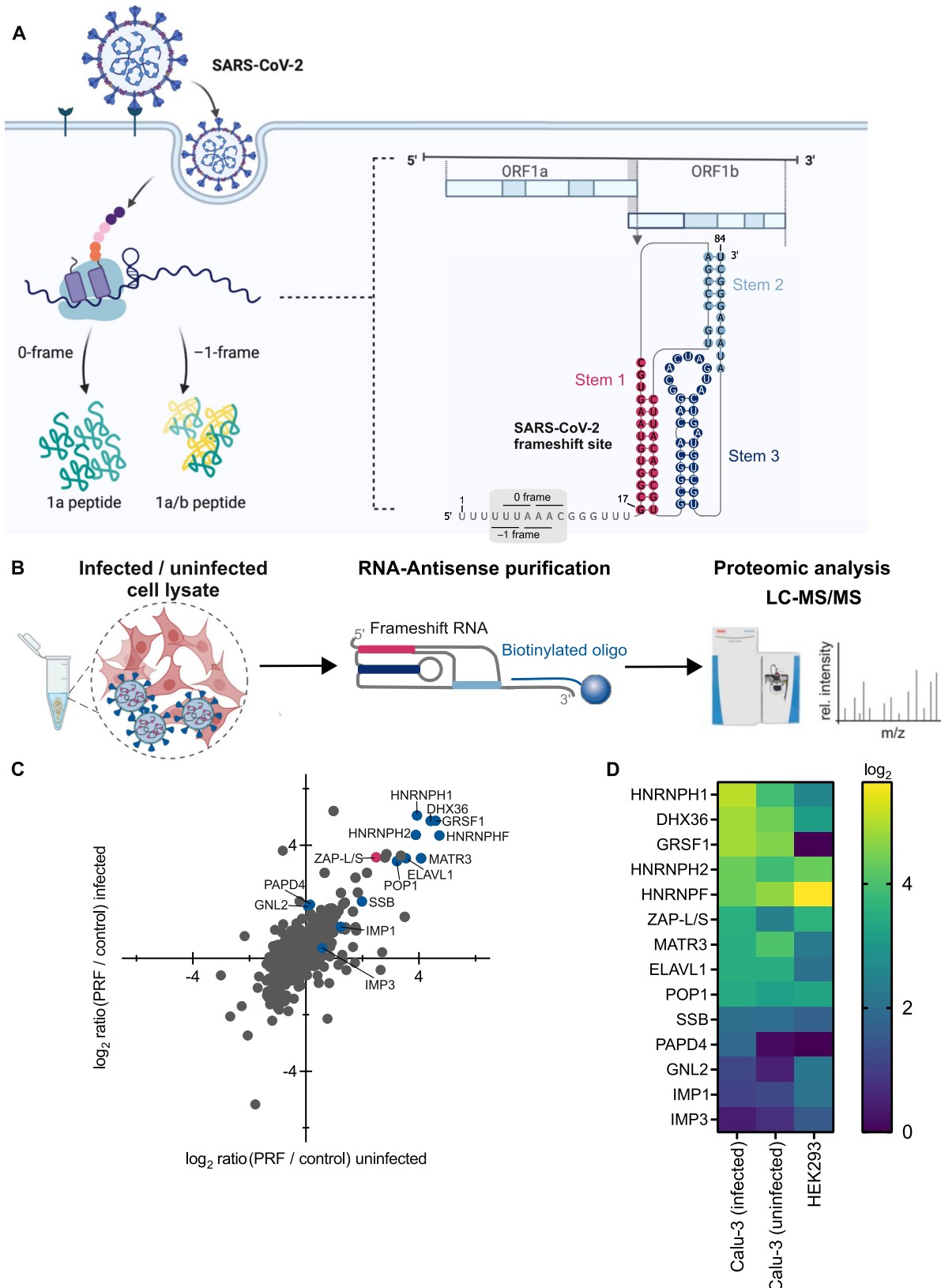

ELAVL1, DHX36, SSB, IMP1 and IMP3[27,28]. Among those, ELAVL1 is a cofactor, which ensures translational fidelity in the context of upstream ORFs[29]. IMP1 and IMP3 were both relatively lowly enriched RBPs in infected and uninfected Calu-3 cells. IMP1 was reported predominantly bound to the 3' untranslated region of genes and IMP3 was mostly bound to coding regions[28,30]. DHX36 is a multifunctional helicase and is involved in translation and innate immunity[31,32]. G-rich RNA sequence binding protein, GRSF1, is implicated in mitochondrial translation[33]. Another multifunctional protein that was identified in our screen was ZAP, which is an interferon-induced antiviral factor with two isoforms (ZAP-S and ZAP-L). Both isoforms of

**Fig. 1 In vitro RNA-antisense purification-based discovery of protein interactors of the SARS-CoV-2 –1PRF element. A** Schematic representation of the relevant genomic segment of SARS-CoV-2 as well as the location of the –1PRF element. **B** Schematic of in vitro interactome capture of protein interactors of the SARS-CoV-2 –1PRF element. In vitro synthesized RNA fragment numbered 1–84 corresponding to nucleotides 13456–13570 of the SARS-CoV-2 genome, was incubated with lysates of naïve HEK293 cells as well as SARS-CoV-2-infected and uninfected Calu-3 cells. The –1PRF RNA was captured by a biotinylated antisense DNA oligo and isolated proteins were subjected to LC-MS/MS. **C** Representative scatter plot of log$_2$-ratios comparing proteins captured in uninfected vs. SARS-CoV-2-infected Calu-3 cells. Core interactors common between uninfected and SARS-CoV-2-infected Calu-3 cells as well as uninfected HEK293 cells are highlighted in blue, ZAP is highlighted in pink. **D** Heatmap representing the enrichment (log$_2$) of core interactors. See also Supplementary Fig. 1D.

ZAP are implied in various RNA-related mechanisms, including RNA decay and translation[26,34–37]. While the longer isoform of ZAP (ZAP-L) was reported to be mainly recruited to membrane-associated sites of viral replication[34,38,39], the shorter cytoplasmic form of ZAP (ZAP-S) has been identified as an immune-regulatory protein through its interaction with the 3′ untranslated region of interferon mRNAs[34]. Two additional hits were included in the downstream analysis based on their above fourfold enrichment only in infected Calu-3 lysates. These included the poly-(A) polymerase PAPD4, and GNL2 which has been implied in ribosome biogenesis[40,41].

**RNA interactors specifically inhibit SARS-CoV-2 frameshifting in cells.** To explore the potential role of the RNA binders in SARS-CoV-2 frameshifting, we designed a fluorescence-based cellular –1PRF reporter assay. In this assay, the expression of the first ORF, EGFP in the 0-frame, is constitutive, whereas the expression of the following ORF mCherry depends on –1PRF occurring at the preceding SARS-CoV-2 1a/1b frameshift element (Fig. 2A). As controls, we used a construct lacking the frameshift element, and the mCherry gene placed either in –1 or in-frame with respect to EGFP (Fig. 2A, B). Frameshifting efficiencies were calculated as the ratio of mCherry to EGFP in the test construct normalized to the in-frame control *(see also Materials and Methods)*. In order to study the effect of the *trans*-acting factors on SARS-CoV-2 frameshifting, cells were co-transfected with both the dual-fluorescence reporter plasmid and the plasmid encoding the putative *trans*-factor as an N-terminal ECFP fusion. This allowed gating of ECFP + cells, which express the *trans*-acting protein of interest (see also Supplementary Fig. 2A). To benchmark the assay, a vector expressing only ECFP was used as a control to compensate for the spectral overlap between ECFP and EGFP. Using this fluorescence reporter assay, the FE of the wild type SARS-CoV-2 1a/1b frameshift site was 35% in HEK293 cells (Fig. 2C, Supplementary Fig. 2A, B), in agreement with the published FE for SARS-CoV-1 and SARS-CoV-2[7,8]. In addition, a vector expressing ECFP-SHFL, a previously described inhibitor of –1PRF in SARS-CoV-2, was employed as a positive control (FE was reduced by 27%)[3]. Among the selected RNA interactors, no change in FE was observed with GNL2, HNRNPF, SSB, IMP1 or IMP3, indicating that binding to the stimulatory RNA element alone is not sufficient for modulating FE. Furthermore, control proteins that were not significantly enriched in the interactome capture, such as SART, DDX3, PINX and ZFR, did not lead to significant changes in FE, corroborating the specificity of the flow-cytometry-based frameshifting assay (Fig. 2C). Two hits, namely GRSF1 and PAPD4, led to a small but statistically significant increase in FE. Proteins with the strongest effect on FE were HNRNPH1, HNRNPH2 and ZAP-S, where frameshifting was substantially reduced by up to 50%. Despite equal expression levels (Supplementary Table 2), the large isoform of ZAP (ZAP-L) reduced frameshifting levels to a much lower degree compared to HNRNPH1, HNRNPH2 and ZAP-S.

We also compared the relative mRNA expression levels of the selected RBPs in published RNA-seq datasets from infected Calu-

3, Huh7.5.1 cells and COVID-19 patients (Supplementary Fig. 1E)[42,43]. HNRNPH1 and HNRNPH2 expression levels did not change upon infection, whereas IMP3 and ZAP transcripts were enriched by more than sixfold in patient samples[42]. We therefore decided to include IMP3 as a control RBP for the downstream analysis due to its relatively low enrichment in the screen (log$_2$ enrichment 0.4–0.7) (Fig. 1C). Notably, among all the hits we analyzed, ZAP was the only factor that was also induced in infected Calu-3 and Huh7.5.1 cells. We also analyzed expression levels of these candidates by quantitative RT-PCR in SARS-CoV-2 infected Calu-3 cells compared to uninfected controls at 72 h post-infection. As seen in the RNA-seq data, only ZAP showed a significant (ca. 20-fold) increase in mRNA levels upon infections[42,43] (Supplementary Fig. 1E, F). An increase in ZAP-S protein levels upon SARS-CoV-2 infection was also reported previously[44,45].

Next, to test whether ZAP-S is functionally relevant during SARS-CoV-2 infection, Huh7 cells stably overexpressing ALFA-tagged ZAP-S were infected with SARS-CoV-2. In line with previous reports using RNAi, ZAP-S overexpression reduced viral replication after 24 hours by approximately 20-fold (Fig. 2D, Supplementary Fig. 2D)[21,46]. We further tested whether the addition of interferons had a synergistic effect but observed no further enhancement of the effect of ZAP-S upon treatment with IFN-α2, INF-ß, IFN-ɣ, and IFN-ʎ1 (Fig. 2D, Supplementary Fig. 2C, E). In addition, we also measured the viral N protein levels via immunofluorescence, which is one of the early markers of SARS-CoV-2 infection. Levels of the N protein were also decreased upon ZAP overexpression (Supplementary Fig. 2C, D and E). Taken together, our results showed that ZAP-S has the potential to restrict SARS CoV-2 replication in our cellular system, similar to published results in Calu-3 cells[46]. Based on its strong induction upon infection, inhibition of viral frameshifting and antiviral function, we decided to focus on ZAP-S for further experiments.

To investigate the specificity of ZAP-S for the SARS-CoV-2 frameshift element, we tested whether the overexpression of ZAP-S affects –1PRF of other RNAs, e.g., different Coronaviruses (SARS-CoV-1, MERS-CoV, Bat-CoV-273, two additional human coronavirus HKU1 and OC43), Arboviruses (West Nile Virus (WNV), Japanese Encephalitis Virus (JEV), Chikungunya Virus (CHIKV)), and Human Immunodeficiency Virus-1 (HIV-1). Our analysis also included the embryonic gene PEG10, which represents an established example for –1PRF in humans[47]. Among the frameshift sites investigated, only the FE of SARS-CoV-1 was reduced significantly in the presence of ZAP-S (decrease by ca. 50%) (Fig. 2E), likely due to the high degree of similarity between the SARS-CoV-1 and CoV-2 frameshift sites. This specificity is unlike the SHFL protein, which affects several PRF genes, including the cellular PEG10[16,48].

In order to understand if the inhibitory effect of ZAP-S on viral frameshifting is dependent on specific interactions with the SARS-CoV-2 frameshift element, we introduced sequential truncations within the predicted stem loops (SL) of the SARS-CoV-2 frameshift stimulatory PK. We prepared a series of

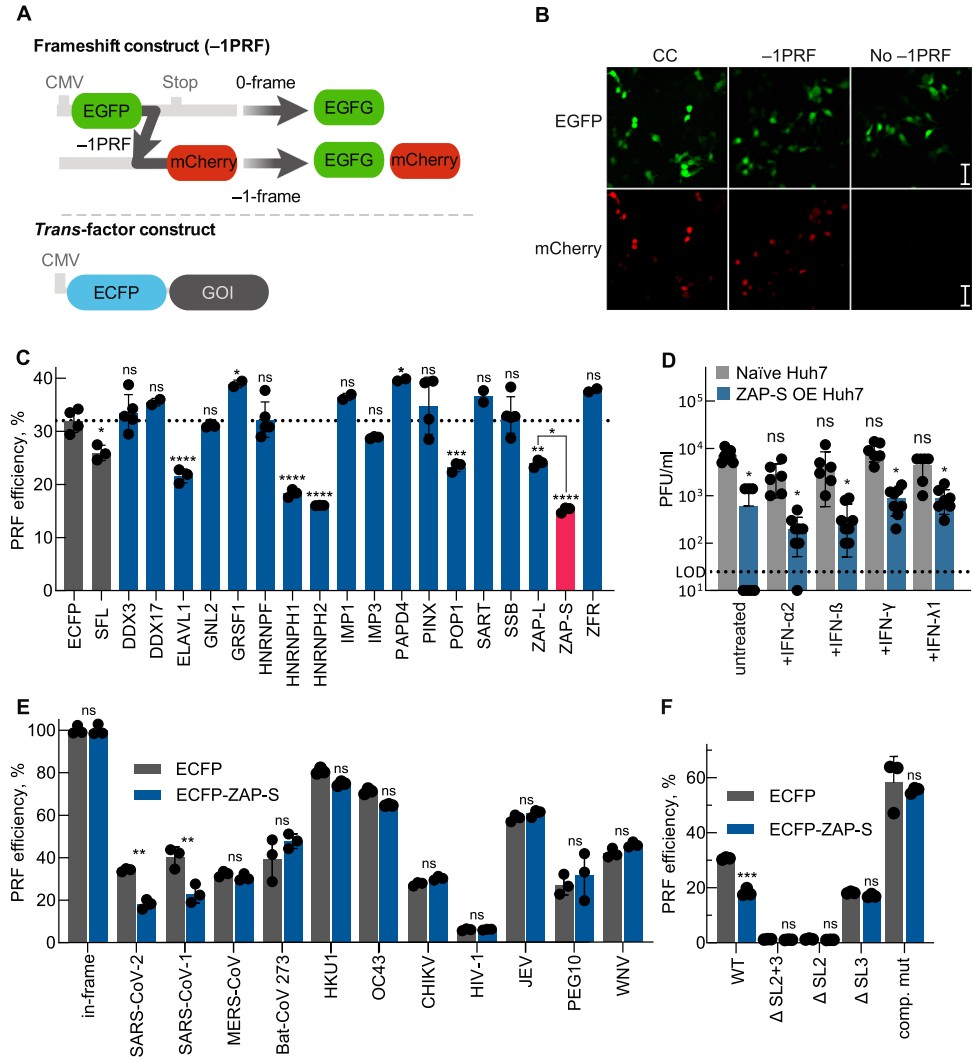

**Fig. 2 A functional screen of SARS-CoV-2 –1PRF element interactors. A** Schematic representation of the dual-fluorescence frameshift reporter construct. EGFP and mCherry are separated by a self-cleaving 2 A peptide as well as by a stop codon in-frame with EGFP. As a result, 0-frame translation would produce only EGFP, whereas –1PRF would produce both EGFP and mCherry. The ratio of mCherry to EGFP fluorescence is used to quantify the FE. The *trans*-factor construct is an N-terminal fusion of ECFP with the protein of interest to be analyzed. The control construct consists of ECFP alone. **B** Confocal microscopy images of cells transfected with the EGFP-mCherry control (CC- no –1PRF site included after EGFP and mCherry in-frame with EGFP), –1PRF, and no PRF (no –1PRF site and stop codon after EGFP) constructs. The size bar represents 50 μm. $n = 1$ independent experiment. **C** Comparison of relative FE of cells overexpressing *trans*-factors as ECFP fusion proteins. Data points represent the mean ± s.d. ($n = 3$ independent experiments). P values were calculated using an ordinary unpaired one-sided ANOVA comparing every condition to the ECFP control. ZAP-L and ZAP-S were separately compared to each other. *$P < 0.05$, **$P < 0.01$, ***$P < 0.001$, ****$P < 0.0001$. Exact P values: SFL – 0.03, DDX3 – 0.99, DDX17 – 0.72, ELAVL1 – < 0.0001, GNL2 – 0.99, GRSF – 0.03, HNRNPF – 0.99, HNRNPH1 – < 0.0001, HNRNPH2 – < 0.0001, IMP1 – 0.39, IMP3 – 0.68, PAPD4 – 0.01, PINX – 0.74, POP1 – 0.0005, SART – 0.36, SSB – 0.99, ZAP-L – 0.001, ZAP-S – < 0.0001, ZFR – 0.12. **D** Virus titers in the supernatant of infected naïve Huh7 or ZAP-S overexpressing Huh7 cells (ZAP-S OE) at 24 h post infection. Treatment with IFN-γ (500 U/ml), IFN-β (500 U/ml), or IFN-λ1 (5 ng/ml) was done 1 h before infection. Boxes show mean values ± s.d. ($n = 4$ independent experiments). The dotted line represents the limit of detection (LOD). P values were calculated using an ordinary unpaired one-sided ANOVA comparing every condition to untreated naïve infected Huh7 cells. Exact P values: untreated +ZAP-S – 0.01, INF-α2 + ZAP-S – 0.04, INF-ß + ZAP-S – 0.49, INF-γ + ZAP-S – 0.049, INF-λ + ZAP-S – 0.049. **E** In vivo dual-fluorescence of additional –1PRF RNAs in HEK293 cells in the presence and absence of ZAP-S. SARS-CoV-1 – severe acute respiratory syndrome-related coronavirus 1, MERS-CoV – Middle East respiratory syndrome-related coronavirus, Bat-CoV-273 – Bat Coronavirus 273, HKU1 – Human coronavirus *HKU1*, OC43 – Human Coronavirus *OC43*, CHIKV – Chikungunya Virus, HIV-1 – Human Immunodeficiency Virus 1, JEV – Japanese Encephalitis Virus, PEG10 – paternally expressed 10, WNV – West Nile Virus. Data points represent the mean ± s.d. ($n = 3$ independent experiments). P values were calculated using an ordinary unpaired one-sided ANOVA comparing every condition to the ECFP control. *$P < 0.05$, **$P < 0.01$. Exact P values: SARS-CoV-2 – 0.001, SARS-CoV-1 – 0.001. **F** In vivo dual-fluorescence of mutants of SARS-CoV-2 –1PRF RNA in HEK293 cells in the presence and absence of ZAP-S. Datapoints represent the mean ± s.d. ($n = 3$ independent experiments). P values were calculated using an unpaired one-sided ANOVA comparing values of the ECFP control. * $P < 0.05$. Exact P values: WT – 0.0003. See also Supplementary Table 2 as well as Fig. 4 for schematics of the mutants used here.

mutants – namely ΔSL2, ΔSL3 and ΔSL2 + 3 – which were deletions of the predicted SL2 region (nucleotides 13535-13542), SL3 region (nucleotides 13505-13532) and both SL2 and SL3 (nucleotides 13505-13542), respectively. Frameshifting was completely abolished in the ΔSL2 and ΔSL2 + 3 mutants, which is in line with minimal sequence requirements for frameshifting in other coronaviruses (Fig. 2F)[9,49]. Due to the absence of PRF in ΔSL2 and ΔSL2 + 3, we were not able to evaluate the effect of ZAP-S with these mutants. With the ΔSL3 mutant, FE was severely reduced (to ~20%) and remained unaffected by the presence of ZAP-S. ZAP has been shown to bind CG dinucleotides[46]. Therefore, we tried to address four of these by compensatory mutants which would maintain the predicted base pairing. These compensatory C<–>G mutations led to an increase of the FE up to 60%, which might be due to stabilization of the PK or alternatively due to effects on alternative folds. Notably, the PRF-inhibitory effect of ZAP-S was no longer observed in this compensatory mutant (Fig. 2F). Taken together, ZAP-S seems to require an intact PK sequence or a particular RNA fold for its effect, since mutations or truncations in the RNA either decreased or completely abolished its effect.

**ZAP-S decreases SARS-CoV-2 frameshifting efficiency in vitro.** We next focused on characterizing ZAP-S mediated regulation of frameshifting in vitro using the rabbit reticulocyte lysate (RRL) translation system and recombinant ZAP-S. (Fig. 3A and Supplementary Fig. 2F). We employed reporter mRNAs containing a flag tag followed by nucleotides 12686-14190 of the SARS-CoV-2 genome to best mimic the native genomic context of viral frameshifting. Control RNAs exclusively producing either the 0-frame (nsp9-11) or –1-frame products (nsp9-11 + partial nsp12) were employed as size markers for the western blot (Fig. 3B). In accordance with a previous study[7], SARS-CoV-2 FE was about 46% in the absence of ZAP-S. Upon titration of increasing amounts of ZAP-S, we observed a corresponding decrease in FE. At the highest concentration of ZAP-S (3 µM), FE was reduced from 46 to ~26% (Fig. 3B, C). These results establish that ZAP-S acts on the native SARS-CoV-2 mRNA directly and that no cofactors are required for its action. To ensure that the observed effect was specific to ZAP-S and not mediated by non-specific RNA-protein interactions, we also tested IMP3, an RBP that we identified as a weak interactor with the RNA frameshifting element in our screen, and the SUMO-tag alone. Neither the addition of IMP3, nor the addition of SUMO alone led to a change in frameshifting levels (Fig. 3C).

Several *trans*-acting factors including the cardiovirus 2 A and SHFL were shown to bind to ribosomes and as well as frameshifting RNAs[16,50]. Thus, to explore whether ZAP-S interacts with the translation machinery, we performed polysome profiling of the RRL translating the SARS-CoV-2 frameshift reporter mRNA in presence and absence of ZAP-S (Fig. 3A). Both polysome profiles were similar, suggesting that ZAP-S does not significantly change bulk translation in RRL. In addition, ZAP-S was detected in the monosome (80 S) as well as the polysome fractions; the latter represent the actively translating pool of ribosomes (Fig. 3D). To confirm that the interactions of ZAP-S with ribosomal subunits and polysomes also occurs within cells, we conducted polysome profiling of HEK293 cells overexpressing ZAP-S (Fig. 3A). Also in that case, ZAP-S was detected in ribosomal fractions, including polysomes. In this experimental set-up, we could also detect endogenous ZAP-L in free RNA fractions and to a small extent in ribosome fractions (Fig. 3E). Similar polysome profiles were obtained with cells overexpressing SHFL, which as a known ribosome interactor acts as positive control[16]. We further confirmed that endogenous ZAP-S also

associates with ribosomes in naïve Calu-3 cells via ribosome pelleting (Fig. 3A, F). Collectively, these results indicate that ZAP-S associates with ribosomes, either directly, or indirectly through its interactions with the SARS-CoV-2 mRNA.

**ZAP-S directly interacts with the SARS-CoV-2 frameshift motif.** In order to further dissect the interplay between the SARS-CoV-2 frameshifting RNA and ZAP-S, we performed RNA-protein binding assays using the highly sensitive microscale thermophoresis assay (MST) (Fig. 4A, B). The wild type (WT) PK, derived from nucleotides 13456-13570 of the SARS-CoV-2 genome, was in vitro transcribed and Cy5-labeled at the 3′ end. We also tested the stem-loop truncation variants we designed earlier and stem-loop mutants of the stimulatory PK.

For the wild type SARS-CoV-2 PK, we observed that ZAP-S interaction occurs with a high affinity ($K_D = 110 \pm 9$ nM) (Fig. 4C) indicating that ZAP-S is a direct interaction partner of the frameshift signal. Next, with the ΔSL2 mutant, we detected a weak interaction with ZAP-S which was characterized by a $K_D$ of $672 \pm 164$ nM (Fig. 4D). In contrast, deletion of the SL3 region (ΔSL3) only marginally reduced the affinity of ZAP binding ($K_D = 175 \pm 64$ nM) (Fig. 4E). On the other hand, deletion of both SL2 and SL3 (ΔSL2 + 3), which is predicted to fold into a short stem-loop (SL1) completely abolished ZAP-S binding (Fig. 4F). In contrast, ZAP-S binds to the compensatory mutant, with an affinity close to WT RNA ($K_D = 128 \pm 29$ nM) (Fig. 4G). A negative control RNA with the same nucleotide composition as the WT PRF site but a disrupted PK RNA fold did not bind ZAP-S (Fig. 4H). Furthermore, we tested the binding of two control proteins, IMP3 and SUMO, to the SARS-CoV-2 frameshift motif. Compared to ZAP-S, IMP3 showed an almost 7-fold lower affinity to the RNA ($K_D = 806 \pm 252$ nM). No interaction between SUMO and the frameshift element was detected (Supplementary Fig. 3G). Based on these data, we hypothesized that ZAP-S has multiple binding sites in the putative SL2 and SL3 regions of the PK. We then carried out electrophoretic mobility shift assays (EMSAs), which confirmed multiple binding events on the WT PK RNA, but none with the RNA variant lacking the SL2 and SL3 regions (ΔSL2 + 3) (Supplementary Fig. 3H, I). To further analyze potential changes in the SARS-CoV-2 RNA structure in the presence of ZAP-S we carried out dimethyl sulfate (DMS) mutational profiling with sequencing (DMS-MaPseq) (Supplementary Fig. 4). In the absence of ZAP-S, DMS reactivities were consistent with a significant proportion of the RNA folding into a PK conformation (Supplementary Fig. 4). In the presence of ZAP-S, we witnessed decreases in DMS reactivities in both the loop regions of SL2 and SL3, as well as increases in reactivities in the stems of SL1 and SL2. Overall, our MST and DMS-MaPseq analysis suggest SL2 and SL3 as the main binding sites for ZAP-S.

**ZAP-S prevents the refolding of the stimulatory RNA.** Since ZAP-S directly interacted with the frameshift element, we next tested whether this binding alters the RNA structure and/or mechanical stability of the RNA using single-molecule optical-tweezers assays. To this end, an RNA containing the 68 nucleotides long wild type SARS-CoV-2 PK (nucleotides 13475–13542 of SARS-CoV-2 genome) was hybridized to DNA handles and immobilized on polystyrene beads. We employed exclusively the sequence corresponding to the putative PK to preclude the formation of alternative conformers[49,51–53]. We used the force-ramp method to probe the forces required for (un)folding of the RNA in the presence and absence of ZAP-S. Briefly, the frameshift RNA was gradually stretched at a constant rate, and then the applied force was released while recording the molecular end-to-end extension distances. This allows the RNA molecule to

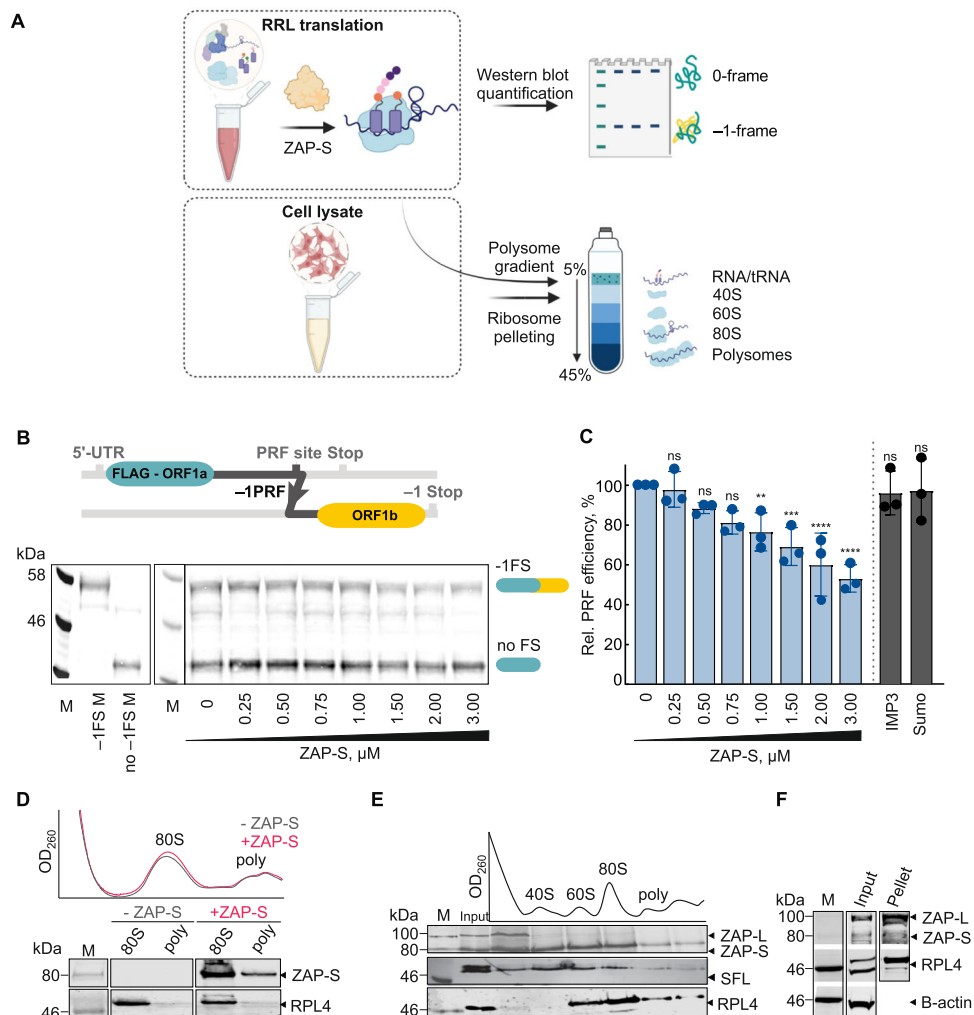

**Fig. 3 Effect of zinc-finger antiviral protein (ZAP) on 1a/1b –1 frameshifting in vitro. A** The strategy of the in vitro translation assay using rabbit reticulocyte lysate (RRL) and the experimental workflow to study ribosome association of ZAP-S. **B** Schematics of the N-terminal FLAG-tagged frameshifting reporter consisting of the nucleotides 12686-14190 (~1.5 kb) of the SARS-CoV-2 genome. RNAs were translated in RRL in the presence of increasing concentrations of ZAP-S ranging from 0 to 3 µM. FLAG-tagged peptides generated by ribosomes that do not frameshift (no –1PRF) or that enter the −1 reading frame (−1PRF) were identified via western blotting using anti-DDDDK antibody. FE was calculated as previously described[11], by the formula: Intensity (–1-frame)/ (Intensity (–1-frame) + Intensity (0-frame)). Size markers - M (Marker), –1PRF M (–1-frame marker), and no –1PRF M (0-frame marker). $n = 3$ independent experiments. **C** Changes in FE observed in the presence of ZAP-S from (**B**) (normalized to 0 µM ZAP as shown in **B**). P values were calculated using an ordinary unpaired one-sided ANOVA comparing every concentration to the no ZAP control. *$P < 0.05$, **$P < 0.01$, ***$P < 0.001$, ****$P < 0.0001$. Exact P values: 0.25 µM – 0.82, 0.50 µM – 0.26, 0.75 µM – 0.06, 1.00 µM – 0.009, 1.50 µM – 0.0002, 2.00 µM –<0.0001, 3.00 µM–< 0.0001. See also Supplementary Fig. 2 and Supplementary Table 3. **D** Polysome profiling analysis of ZAP-S in RRL. RRL translating the FLAG-tagged SARS-CoV-2 frameshifting reporter was subjected to 5–45% sucrose gradient ultracentrifugation, and subsequently fractionated. Levels of RPL4, as well as ZAP in each fraction, were analyzed by western blotting using anti-RPL4 and anti-ZC3HAV1 (ZAP) antibodies. $n = 2$ independent experiments. **E** Ribosome pelleting of untreated Calu-3 cells. Naïve Calu-3 cells were lysed and loaded onto sucrose cushions. Levels of RPL4, ZAP, and β-actin in the pellets were analyzed by western blotting using anti-RPL4, anti-ZC3HAV1 (ZAP) and anti-β-actin antibodies. $n = 3$ independent experiments. **F** Polysome profiling analysis of ZAP-S in cells. HEK293 cells transiently expressing ZAP-S were lysed, subjected to 5–45% sucrose gradient ultracentrifugation, and subsequently fractionated. Levels of ribosomal proteins, ZAP as well as SHFL in each fraction, were analyzed by western blotting using anti-RPL4, anti-ZC3HAV1 (ZAP) and anti-RYDEN (SHFL) antibodies. $n = 3$ independent experiments.

transition between folded and unfolded states, and sudden changes in measured force-distance trajectories indicate transitions between various RNA conformations (Fig. 5A). By mathematically fitting each force-distance trajectory, we can obtain information on the physical properties of the RNA such as the change in the contour length (number of nucleotides unfolded) or the force required for (un)folding (*Methods*). With the SARS-CoV-2 putative PK, in the absence of ZAP-S, we mainly observed a single-step unfolding event leading to a contour length ($L_C$) change of $35.4 \pm 3.0$ nm (Supplementary Fig. 6, Supplementary

Table 1), which agreed with the expected value for the full-length PK reported previously Fig. 5B)[51,52]. Moreover, the majority (80%) of RNA molecules unfolded at forces ($F_U$) of 15–20 pN (Supplementary Fig. 5). For the remaining traces, we observed two consecutive unfolding events with an intermediate contour length change of $17.1 \pm 3.5$ nm (Fig. 5D) likely corresponding to the sequential unfolding of the PK structure. By decreasing the force, the RNA refolded in two steps, both at about 11 pN (Fig. 5D, Supplementary Fig. 6, Supplementary Table 1). Such a hysteresis during refolding is commonly reported with PKs and

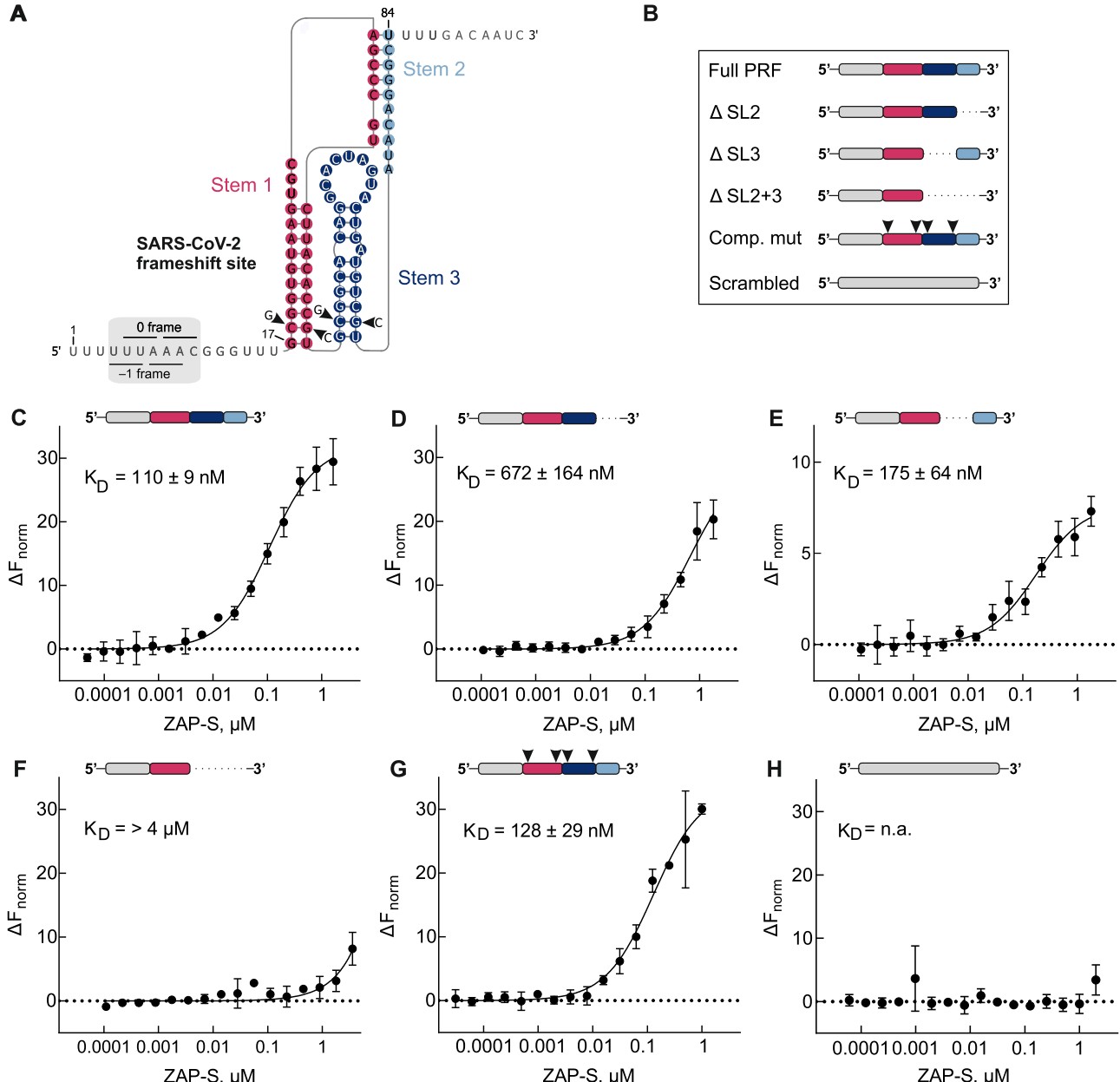

**Fig. 4 In vitro characterization of ZAP-S interaction with SARS-CoV-2 −1 PRF RNA. A** Proposed structure of the PRF element of SARS-CoV-2. Nucleotide substitutions in the compensatory mutant are indicated (arrowheads). **B** Schematic representations of the RNAs studied. **C**–**H** Microscale thermophoresis assay to monitor ZAP-S binding to (**C**) Full PRF, (**D**) ΔSL2 mutant, (**E**) ΔSL3 mutant, (**F**) ΔSL2 + 3 mutant, (**G**) compensatory mutant, (**H**) scrambled mutant. Unlabeled protein (40 pM–2 µM) was titrated against 3′ pCp-Cy5 labeled RNA (5 nM) and thermophoresis was recorded at 25 °C with 5% LED intensity and medium MST power. Change in fluorescence ($\Delta F_{norm}$) was measured at MST on-time of 2.5 s. Data were analyzed for $\Delta F_{norm}$ using standard functions of MO. Affinity Analysis software and data was plotted and $K_D$ was determined using Graphpad Prism 9.2.0. Data represent mean ± s.d. of three measurements ($n = 3$). For the related thermophoretic traces, see also Supplementary Fig. 4A-F. For the related DNA sequences of the mutants, see also Supplementary Table 2.

other highly structured RNAs[51,52]. When we performed the measurements in the presence of ZAP-S RNA unfolding trajectories remained almost/mostly unaffected, suggesting that the interaction neither stabilizes nor destabilizes the RNA structure (Fig. 5D, Supplementary Figs. 5 and 6, Supplementary Table 1). On the other hand, strikingly, refolding of the RNA into its native fold was impaired with less or no detectable transitions into the folded state (Fig. 5D, Supplementary Figs. 5 and 6, Supplementary Table 1).

To better characterize the sequence or structural constraints that are important for the ZAP-S mediated effect, we also employed the same set of truncation mutants of the wild type SARS-CoV-2 PK used earlier (Figs. 4B, 5C E–H). Truncation of SL2 region (ΔSL2) is expected to prevent the formation of the PK, and instead RNA would fold into two consecutive SL (Fig. 5C, E). With ΔSL2 both the change in $L_C$ (30.8 ± 3.1 nm) and $F_U$ (peak 1 − 9.3 ± 1.3 pN, peak 2–13.8 ± 0.8 pN) were lower compared to the wild type PK, and RNA was able to refold back readily, which was in line with the formation of predicted stem-loops. In the presence of ZAP-S with the ΔSL2 variant, force of unfolding was unchanged, but three distinct populations of refolding were observed based on the change in the contour length

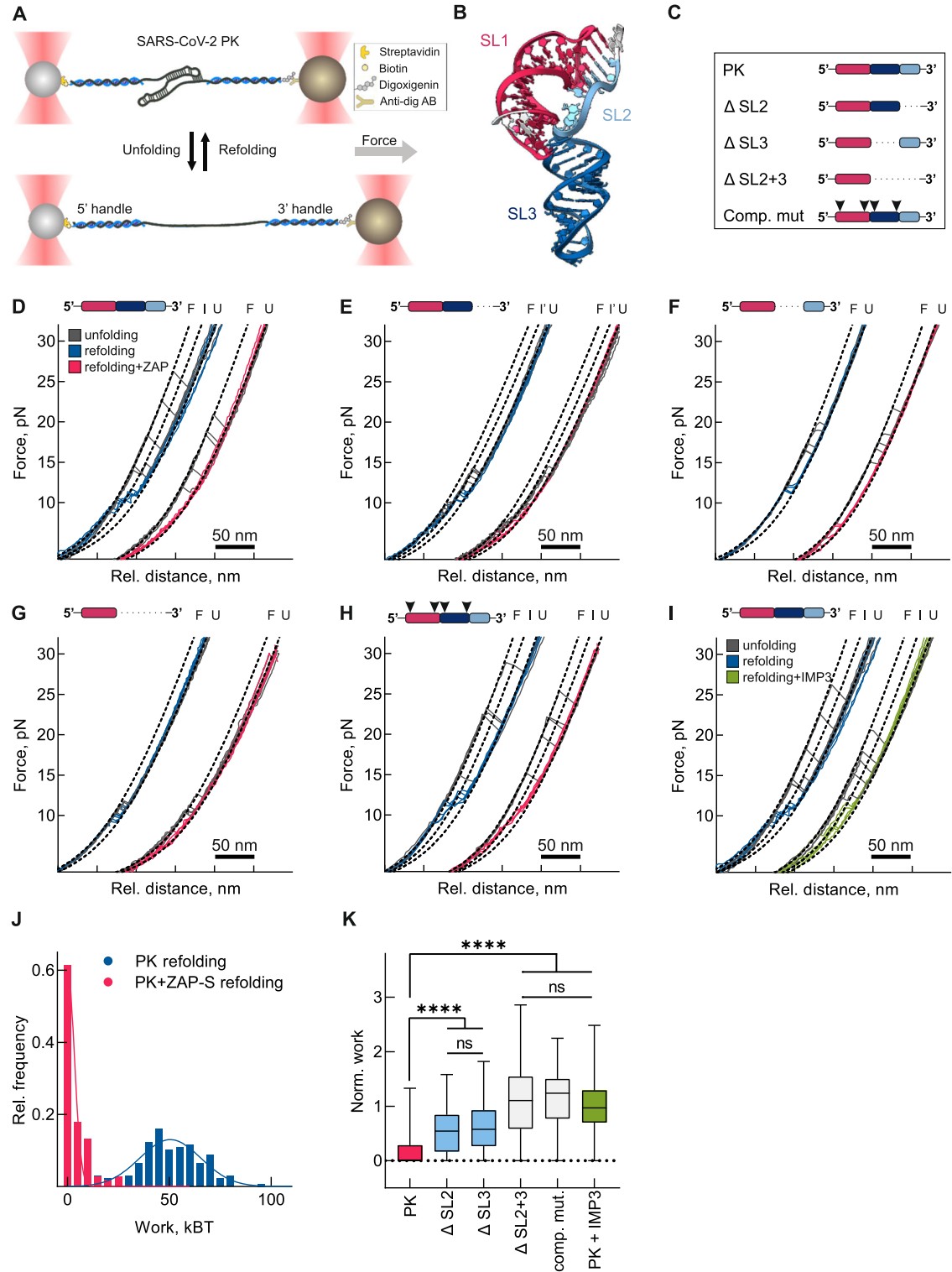

(Supplementary Figs. 5 and 6, Supplementary Table 1). In one population no refolding was seen (0.2 ± 0.3 nm), the second one showed similar step sizes during (un)folding (25.6 ± 2.8 nm), and the third one represented a partially refolded state, which was likely a simple hairpin based on the lower contour length change (15.9 ± 3.1 nm) (Supplementary Fig. 6, Supplementary Table 1). In the ΔSL3 RNA variant, (Fig. 5C, F), the RNA was predicted to fold into a shorter PK. In agreement with this prediction, we measured higher forces of unfolding (17.4 ± 1.3 pN) and hysteresis during refolding, yet the change in contour length

(21.0 ± 1.2 nm) was lower than the wild type PK (Supplementary Figs. 5 and 6, Supplementary Table 1). In the presence of ZAP-S, no refolding was observed in about 20% of ΔSL3 curves, and we observed a significant decrease in the refolding work (Fig. 5F, Supplementary Table 1). The ΔSL2 + 3 variant is predicted to form the simple hairpin (SL1). Our data also confirmed the presence of a single stem-loop (Fig. 5G, Supplementary Fig. 5 and 6, Supplementary Table 1), with the contour length value of (16.4 ± 2.8 nm). Here, only about 10% of traces did not refold in the presence of ZAP-S (Fig. 5G). Aside from that, $F_U$ was slightly

**Fig. 5 Single molecule characterization of mechanical properties of SARS-CoV-2 PRF RNA in the presence of ZAP-S. A** Schematic illustrating optical tweezers experiments. RNA was hybridized to single-stranded DNA handles flanking the SARS-CoV-2 frameshift site and conjugated to functionalized beads. A focused laser beam was used to exert pulling force from one end of the molecule. The force was gradually increased until the RNA was fully unfolded (bottom). **B** 3D structure of SARS-CoV-2 pseudoknot RNA (PK) derived from[51] and colored according to the scheme used in Fig. 4. **C** Schematic representations of the RNAs studied. **D–I** Example unfolding and refolding traces of PK in the presence or absence of ZAP-S, "F" denotes the folded state, "I" the intermediate, and "U" the fully unfolded state, (**D**) PK ($N = 273$ FD curves from 24 molecules no ZAP-S, $N = 219$ FD curves from 24 molecules +ZAP-S samples), (**E**) ΔSL2 mutant ($N = 146$ FD curves from eight molecules no ZAP-S, $N = 122$ FD curves from eight molecules +ZAP-S samples), (**F**) ΔSL3 mutant ($N = 127$ FD curves from 12 molecules no ZAP-S, $N = 163$ FD curves from 11 molecules +ZAP-S samples), (**G**) ΔSL2 + 3 mutant ($N = 216$ FD curves from eight molecules no ZAP-S, $N = 196$ FD curves from 11 molecules +ZAP-S samples), (**H**) compensatory mutant ($N = 158$ FD curves from 12 molecules no ZAP-S, $N = 169$ FD curves from 16 molecules +ZAP-S samples), (**I**) PK in absence (blue) and presence (green) of IMP3 ($N = 273$ FD curves from 24 molecules no ZAP-S, $N = 226$ FD curves from 20 molecules +ZAP-S samples). **J** Distribution of refolding work in presence (pink) and absence (blue) of ZAP-S. **K** Normalized refolding work in the presence of ZAP-S or IMP3. Data points represent the mean ± s.d. (box) and min and max values (whiskers). *P* values were calculated using an ordinary unpaired one-sided ANOVA followed by Dunnett's multiple comparisons test. * $P < 0.05$ – **** $P < 0.00001$. See also Supplementary Figs. 5, 6 and Supplementary Table 1.

shifted to lower values although our MST results clearly showed no binding of ZAP-S to this RNA variant (Fig. 4F, Supplementary Fig. 5). We also tested the effect of non-specific interactions using the control RBP IMP3, and we observed a similar small shift in the $F_U$. Therefore, we conclude that this effect is due to non-specific interactions and/or molecular crowding (Fig. 5I). Finally, with the compensatory mutant (comp. mut.), which has a stack of 4Gs at the SL1 and SL3, unfolding forces were slightly higher than with the WT PK (18.9 ± 5.5 pN). Nevertheless, the contour length change matched with the expected PK structure (36.3 ± 1.7 nm) (Fig. 5H, Supplementary Figs. 5 and 6, Supplementary Table 1). While we cannot exclude that the compensatory mutant forms an alternative structure to the wild type PK, we hypothesize that this stabilization might be caused by the stacking interactions between G stretches at the base of the stems. Interestingly, force-extension behavior of this alternative PK was only minimally affected by ZAP-S binding (Fig. 5H).

To further compare the effect of ZAP-S on SARS-CoV-2 RNA variants, we calculated the work performed during refolding of the RNAs in the presence and absence of ZAP-S (Fig. 5J, Supplementary Table 1). Since work is calculated as a numerical integration of FD curves (Methods), employing of the refolding work enabled us to account for the ZAP-S effect on both refolding force as well as the total contour length change in a single value, thus allowing a better comparison among different samples. In the wild type PK work performed during refolding in the presence of ZAP-S was negligible, and the majority of traces (more than 60%) do not show any detectable refolding. Since the other RNAs differed in their lengths and other physical properties, we normalized the refolding work performed on each RNA in the presence of ZAP-S to work performed in the absence of ZAP-S. This allowed a non-biased comparison of the effect of the *trans*-acting factor. No significant difference in work was detected with PK in the presence of IMP3 control, or ΔSL2 + 3 and comp. mut. in the presence of ZAP-S (Fig. 5K). Conversely, in ΔSL2 and ΔSL3 RNA variants the refolding work was still affected by ZAP-S, albeit to a lesser degree when compared to wild type PK. Overall, we were able to quantify the effect of ZAP-S on refolding of the PK RNA and we suggest that SL2 and SL3 are crucial for the function of ZAP-S.

## Discussion

Programmed ribosomal frameshifting (−1PRF) is essential for coronavirus replication. In this study, we explored whether *trans*-acting host or viral factors can modulate SARS-CoV-2 −1PRF. We discovered that the short isoform of the interferon-induced zinc-finger antiviral protein ZAP-S can strongly impair SARS-CoV-2 frameshifting and decrease viral replication. ZAP-S was

also one of the prominent common hits in genome-wide screens for proteins that interacted with SARS-CoV-2 RNA[3,21–23].

Similar to previously reported *trans*-acting protein regulators of frameshifting (such as cardiovirus 2 A and SHFL), we show that ZAP-S interacts with the translation machinery, suggesting that this might be a common feature of PRF modulators. Yet, unlike the cardiovirus 2 A or cellular poly(C)-binding protein, ZAP-S does not mediate formation of a more stable mRNA roadblock to induce frameshifting[12,15]. Rather, ZAP-S inhibits coronaviral frameshifting through recognition of a specific RNA motif. This sequence preference is not common among frameshift modulators. For example, SHFL interacts with stalled ribosomes and recruits release factors to terminate translation irrespective of the type of frameshift RNA[16]. In contrast, ZAP-S shows sequence preference for SARS-CoV-2 SL2 and SL3. We suggest that ZAP-S has multiple binding modes and binds to folding intermediate structures, which includes SL3, which we showed was essential for PRF inhibition. ZAP-S binding may thus slow down the folding rate of the native PK structure. Interestingly, SL3 is identical in SARS-CoV-1 and −2 frameshift motifs but shows a higher degree of sequence diversity in other coronaviruses. Furthermore, the compensatory mutant with a stretch of four Gs at the base of SL1 and SL3 was the most effective in stimulating PRF. The high FE of this mutant can be explained by the thermodynamic stability of the first 3-4 base pairs of the SL, near the mRNA entry channel of the ribosome[54]. Strikingly, the *trans*-acting factor ZAP-S showed no strong effect on this RNA variant, even though it interacts with the RNA element in the steady state. Here the effect of ZAP-S might not be prominent, either because the binding site or the structure is somehow altered due to the mutations or due to faster refolding kinetics of the PRF stimulatory element. This supports the notion that binding of ZAP-S is a prerequisite but not sufficient for its modulatory effect. Furthermore, it may explain why not all binders identified in our screen or in other studies are affecting frameshifting levels.

Ultimately, based on our findings, we propose the following model for the inhibition of −1PRF by ZAP-S (Fig. 6). ZAP-S binding to the frameshift RNA alters the stimulatory RNA structure and reduces the chance of elongating ribosomes to encounter the stimulatory PK. Without this stimulatory structure, the elongation pause during the next round of translation would be too short for codon-anti-codon interactions to be established in the −1-frame. Thus, ZAP-S would likely allow translation to proceed and terminate at the 0-frame UAA stop codon found immediately downstream of the slippery sequence. The resulting decrease in the amounts of the 1a/1b polypeptides may lead to a reduction in the levels of the viral RNA-dependent RNA polymerase (RdRP) from the −1-frame.

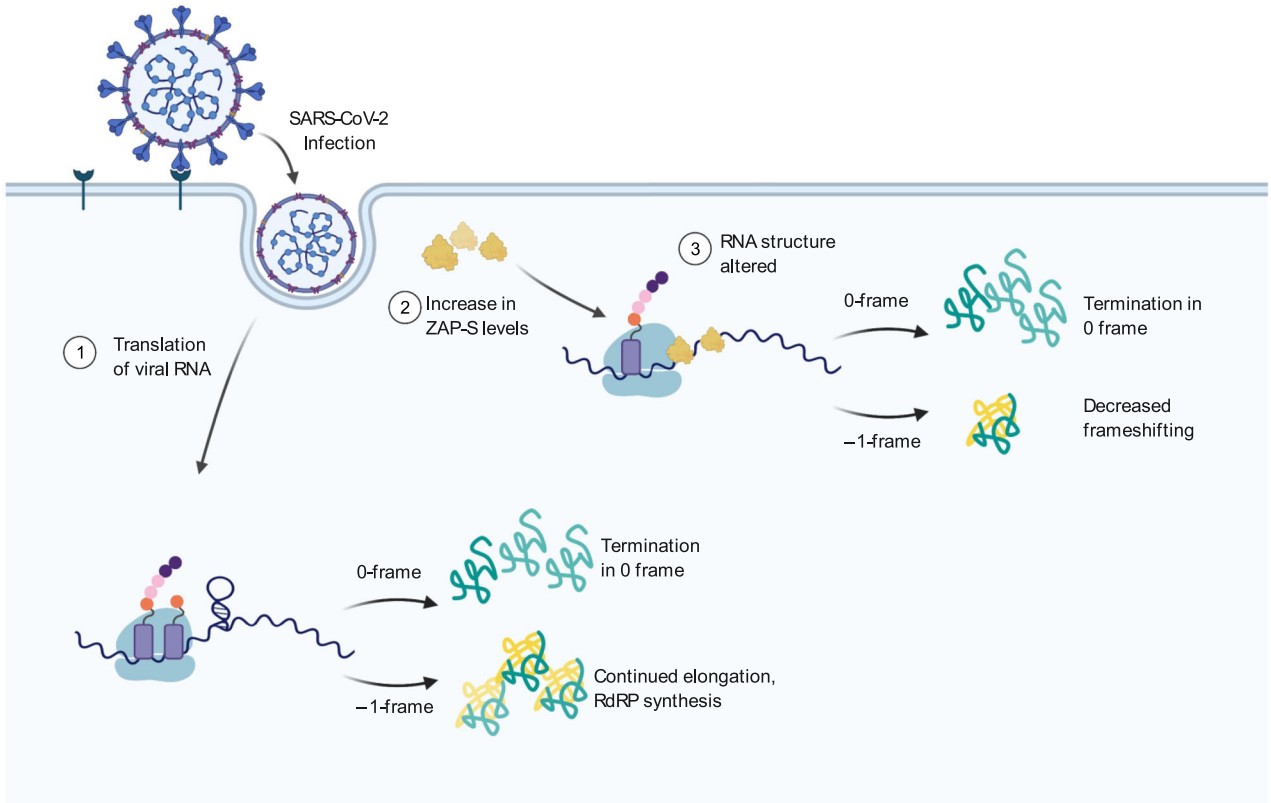

**Fig. 6 Model for ZAP-S mediated inhibition of SARS-CoV-2 frameshifting. 1** Upon infection, the viral RNA is translated by the cellular machinery, and 40% of translation events yield the 1a/1b polyprotein through −1PRF. **2** Infection also leads to the induction of antiviral factors including ZAP-S. **3** ZAP-S binding to the frameshift RNA alters RNA refolding and thereby reduces the chance of elongating ribosomes to encounter the stimulatory structure. Thus, the elongation pause is too short for codon-anti-codon interactions to be established in the −1-frame and ZAP-S allows translation to proceed without a strong roadblock effect. This leads to termination at the canonical 0-frame UAA stop codon found just downstream of the slippery sequence. The resulting decrease in the amounts of the 1a/1b polypeptides reduces the levels of the viral RNA-dependent RNA polymerase (RdRP) from the −1-frame.

In addition to the direct interaction with the frameshifting RNA element, ZAP-S also associates with the ribosomes, although how it interacts, direct or indirectly, or whether this interaction is functionally relevant awaits further investigation. We envision ZAP-S binding to the native PK and inhibiting its interactions with the ribosome as one possibility. Overall, these findings establish ZAP-S as unique cellular factor, which has a direct role in modulating SARS-coronavirus frameshifting. In accordance with previously published results, we demonstrate that over-expression of ZAP-S reduces the replication of SARS-CoV-2[21,46]. Further studies are required to deconvolute the multivalent effects of ZAP-S on immunity, viral replication and gene expression[34,55–59]. Given the plethora of mechanisms by which *trans*-regulators of PRF can act, it is conceivable that viral- and host-encoded *trans*-factors follow a multitude of routes to impact frameshift paradigms. Taken together, our study establishes ZAP-S as a novel regulator of SARS-CoV-2 frameshifting and determines one (potential) mechanism by which ZAP-S mediates a SARS-CoV-2 antiviral response.

## Methods

**RNA affinity pulldown mass spectrometry**. RNA antisense purification was performed according to a protocol based on[18]. Briefly, 6*10^7 HEK293 cells per condition were lysed in a buffer containing 20 mM Tris/HCl pH 7.5, 100 mM KCl, 5 mM $MgCl_2$, 1 mM DTT, 0.5% Igepal CA630 (Sigma-Aldrich), 1× complete™ Protease Inhibitor Cocktail (Roche), 40 U/ml RNase inhibitor (Molox). The cleared lysate was incubated with in vitro transcribed RNA corresponding to the SARS-CoV-2 −1PRF site, which was immobilized on SA hydrophilic magnetic beads (NEB) by biotin-SA interaction. After three washes with binding buffer (50 mM HEPES/KOH pH 7.5, 100 mM NaCl, 10 mM $MgCl_2$) and two washes with wash buffer (50 mM HEPES/KOH pH 7.5, 250 mM NaCl, 10 mM $MgCl_2$), bound

proteins were eluted by boiling the sample in 1× NuPAGE LDS sample buffer (Thermo Fisher Scientific) supplemented with 40 mM DTT. For infected as well as uninfected Calu-3 cells the procedure was performed similarly. In order to inactivate the virus, the lysis buffer contained Triton X-100 and inactivation was confirmed by plaque assays.

For LC-MS/MS, the eluted proteins were alkylated using iodoacetamide followed by acetone precipitation. In solution digests were performed in 100 mM ammonium bicarbonate and 6 M urea using Lys-C and after reducing the urea concentration to 4 M with trypsin. Peptides were desalted using C18 stage tips and lyophilized. LC-MS/MS was performed at the RVZ Proteomics Facility (Würzburg, Germany) and analyzed as described previously[60]. Gene ontology (GO) term analysis was performed with Panther[61]. The list of all identified proteins is given in the Source Data File.

**Plasmid construction**. To generate dual-fluorescence reporter constructs frameshift sites of SARS-CoV-1, SARS-CoV-2, MERS-CoV, BtCoV 273, Human coronavirus *HKU1*, Human Coronavirus *OC43*, HIV-1, JEV, PEG10, WNV were placed between the coding sequence of EGFP and mCherry (parental construct was a gift from Andrea Musacchio (Addgene plasmid # 87803[62]) by site-directed mutagenesis or golden gate assembly in a way that EGFP would be produced in 0-frame and mCherry in −1-frame. EGFP and mCherry were separated by StopGo[63] signals as well as an alpha-helical linker[64]. A construct with no PRF insert and mCherry in-frame with EGFP served as a 100% translation control and was used to normalize EGFP and mCherry intensities. Mutants of the frameshift site of SARS-CoV-2 in the dual fluorescence as described in Fig. 4A and Supplementary Table 2 were generated by golden gate assembly.

To generate screening vectors, protein-coding sequences of DD3X (NM_001193416.3), DDX17 (NM_001098504.2), DDX36 (NM_020865.3), ELAVL1 (NM_001419.3), GNL2 (NM_013285.3), GRSF1 (NM_001098477.2), HNRNPF (NM_001098204.2), HNRNPH1 (NM_001364255.2), HNRNPH2 (NM_001032393.2), IGF2BP1 (IMP1) (NM_006546.4), IGF2BP3 (IMP3) (amplified from a vector kindly provided by Dr. Andreas Schlundt), MATR3 iso 2 (NM_018834.6), MMTAG2 (NM_024319.4), NAF1 (NM_138386.3), NHP2 (NM_017838.4), PAPD4 (NM_001114393.3), PINX1 (NM_001284356.2), POP1 (NM_001145860.2), RAP11B (NM_004218.4), RSL1D1 (NM_015659.3), SART1

(NM_005146.5), SHFL (NM_018381.4), SSB (NM_001294145.2), SURF6 (NM_001278942.2), TFRC (NM_003234.4), ZC3HAV1 (ZAP) (NM_024625.4), ZFR (NM_016107.5), and ZNF346 (NM_012279.4) were placed in-frame with the coding sequence for ECFP in pFlp-Bac-to-Mam (gift from Dr. Joop van den Heuvel, HZI, Braunschweig, Germany[65]) via Gibson Assembly.

Golden Gate compatible vectors for heterologous overexpression in *E. coli*, in vitro translation in RRL, and lentivirus production, were generated by Golden Gate or Gibson Assembly. A dropout cassette was included to facilitate the screening of positive colonies. Protein-coding sequences were introduced by Golden Gate Assembly using AarI cut sites[66]. pET-SUMO-GFP (gift from Prof. Utz Fischer, Julius-Maximilians-University, Würzburg, Germany) was used as the parental vectors for protein overexpression in *E. coli*. The lentivirus plasmid was a gift from Prof. Chase Beisel (HIRI-HZI, Würzburg, Germany). An ALFA-tag was included to facilitate the detection of the expressed protein[67]. The frameshift reporter vector for the in vitro translation contained ß-globin 5′ and 3′ UTRs as well as a 30 nt long poly-(A) tail. The insert was derived from nucleotides 12686–14190 of SARS-CoV-2 (NC_045512.2); a 3×FLAG-tag was introduced at the N-terminus to facilitate detection. To generate 0% and 100% −1PRF controls, the −1PRF site was mutated by disrupting the PK structure as well as the slippery sequence.

Optical tweezers (OT) constructs were based on the wild type SARS-CoV-2 frameshift site (nucleotides 13475-13541) cloned into the plasmid pMZ_lambda_OT, which encodes for the optical tweezer handle sequences (2 kb each) flanking the RNA structure (130 nt). Constructs were generated using Gibson Assembly. Sequences of all plasmids and oligos used in this study are given in Source Data File.

**Cell culture, transfections, generation of polyclonal stable cell lines.** HEK293 cells (gift from Prof. Jörg Vogel, HIRI-HZI) and Huh7 cells (gift from Dr. Mathias Munschauer, HIRI-HZI), were maintained in DMEM (Gibco) supplemented with 10% FBS (Gibco) and 100 μg/ml streptomycin and 100 U/ml penicillin. Calu-3 cells (ATCC HTB-55) were cultured in MEM (Sigma) supplemented with 10% FBS. Cell lines were kept at 37 °C with 5% CO₂. Transfections were performed using PEI (Polysciences) according to manufacturer's instructions. For co-transfections, plasmids were mixed at a 1:1 molar ratio.

VSV-G envelope pseudo-typed lentivirus for the generation of stable cell lines was produced by co-transfection of each transfer plasmid with pCMVdR 8.91[68] and pCMV-VSV-G (gift from Prof. Weinberg, Addgene plasmid # 8454[69]). 72 h post-transfection, the supernatant was cleared by centrifugation and filtration. The supernatant was used to transduce naïve Huh7 cells in the presence of 10 μg/ml polybrene (Merck Millipore). After 48 h, the cells were selected with 10 μg/ml blasticidin (Cayman Chemical) for 10 days to generate polyclonal cell lines.

**SARS-CoV-2 infection.** For infection with SARS-CoV-2, we used the strain hCoV-19/Croatia/ZG-297-20/2020, a kind gift of Prof. Alemka Markotic (University Hospital for Infectious Diseases, Zagreb, Croatia). The virus was raised for two passages on Caco-2 cells (HZI Braunschweig). Calu-3 cells (ATCC HTB-55) were infected with 2000 PFU/ml corresponding to an MOI of 0.03 at 24 h post-infection, cells were collected and lysed for proteomic and ribosome-interaction experiments. To study the effect of ZAP-S on SARS-CoV-2 infection, Huh-7 cells were employed. One hour before infection, Huh-7 cells both naïve or ZAP-S-overexpressing cells were either pre-stimulated with IFN-β (500 U/ml), IFN-γ (500 U/ml), IFN-λ1 (5 ng/ml), or left untreated. Cells were infected with 20,000 PFU/ml, corresponding to an MOI of 0.03 at 24 h post-infection, cell culture supernatants were collected and titrated by plaque assay on Vero E6 cells (ATCC CRL-1586). Briefly, confluent Vero E6 cells in 96-well plates were inoculated with dilutions of the virus-containing supernatants for 1 h at 37 °C, the inoculum was removed and cells were overlaid with MEM containing 1.75% methyl-cellulose. At 3 days post-infection, whole wells of the plates were imaged using an IncuCyte S3 (Sartorius) at 4X magnification, and plaques were counted visually.

**Flow cytometry.** HEK293 cells were transiently transfected with either the control construct or the −1PRF construct encoding for the dual-fluorescence EGFP-mCherry translation reporter as outlined in Fig. 2A. Cells were harvested at 24 h post-transfection and fixed with 0.4% formaldehyde in PBS. After washing with PBS, flow cytometry was performed on a FACSAria III (BD Biosciences) or a NovoCyte Quanteon (ACEA) instrument. Flow cytometry data were analyzed with FlowJo software (BD Biosciences). ECFP-positive cells were analyzed for the ratio between mCherry and EGFP (Supplementary Fig. 2F). FE was calculated according to the following formula:

$$FE(\%) = \frac{mCherry_{test}/EGFP_{test}}{mCherry_{control}/EGFP_{test}} \quad (1)$$

where mCherry represents the mean mCherry intensity, EGFP the mean EGFP intensity, test represent the tested sample and control represents the in-frame control where mCherry and EGFP are produced in an equimolar ratio[70]. Data represent the results of at least three independent experiments.

**Purification of recombinant proteins.** Recombinant ZAP-S N-terminally tagged with 6×His-SUMO was purified from *E. coli* Rosetta 2 cells (Merck) by induction with 0.2 mM isopropyl β-d-1-thiogalactopyranoside for 18 h at 18 °C. Cells were collected, resuspended in lysis buffer (50 mM HEPES/KOH pH 7.6, 1 M NaCl, 1 mM DTT, 1 mM PMSF) and lysed in a pressure cell. The lysate was cleared by centrifugation and ZAP-S was captured using Ni-NTA resin (Macherey-Nagel). After elution with 500 mM imidazole, ZAP-S was further purified and the bound nucleic acids removed by size exclusion chromatography (HiLoad® 16/600 Superdex® 200) in 20 mM HEPES/KOH pH 7.6, 1 M KCl, 1 mM DTT, 20% glycerol. Protein identity was verified by SDS-PAGE as well as western blotting (Supplementary Fig. 2D). Purified ZAP-S was rapidly frozen and stored in aliquots at −80 °C. His-SUMO IGF2BP3 as well as His-SUMO were kind gifts from Dr. Andreas Schlundt (Goethe University, Frankfurt, Germany).

**Western blots.** Protein samples were denatured at 95 °C and resolved by 12% SDS-PAGE at 30 mA for 2 h. After transfer using Trans-Blot (Bio-Rad), nitro-cellulose membranes were developed using the following primary antibodies: anti-His-tag (ab18184, dilution 1:1000), anti-DDDDK (ab49763, dilution 1:3000), anti-ALFA (FluoTag®-X2 anti-ALFA AlexaFluor 647, dilution 1:1000), anti-ZC3HAV1 (Proteintech 16820-1-AP, dilution 1:3000), anti-RPL4 (Proteintech 67028-1-Ig, dilution 1:10000), anti-RPS6 (Proteintech 14823-1-AP, dilution 1:500), anti-RYDEN (SHFL; Proteintech 27865-1-AP, dilution 1:1000). The following secondary antibodies were used: IRDye® 800CW Goat anti-rabbit (dilution 1:25000) and IRDye® 680RD Donkey anti-Mouse (dilution 1:15000; both LI-COR). Bands were visualized using an Odyssey Clx infrared imager system (LI-COR) or a Typhoon7000 (GE Healthcare).

**In vitro translation assays.** mRNAs were in vitro transcribed using T7 polymerase purified in-house using linearized plasmid DNA as the template. These mRNAs were capped (Vaccinia Capping System, NEB) and translated using the nuclease-treated rabbit reticulocyte lysate (RRL; Promega). Typical reactions were comprised of 75% v/v RRL, 20 μM amino acids, and were programmed with ~50 μg/ml template mRNA. ZAP-S was buffer exchanged into 250 mM KCl, 50 mM HEPES/KOH pH 7.6, 0.05 mM EDTA, 5% glycerol, 1 mM DTT, Rnasin and titrated in the range of 0–3 μM. Reactions were incubated for 1 h at 30 °C. Samples were mixed with 3× volumes of 1X NuPAGE™ LDS Sample Buffer (Invitrogen), boiled for 3 min, and resolved on a NuPAGE™ 4–12% Bis-Tris polyacrylamide gel (Invitrogen). The products were detected using western blot (method as described above). The nitrocellulose membranes were developed using anti-DDDDK primary (Abcam ab49763) and IRDye® 680RD donkey anti-mouse secondary antibody (LI-COR). Bands were visualized using an Odyssey Clx infrared imager system (LI-COR). Bands corresponding to the −1 or 0-frame products, 58 kDa and 33 kDa respectively, on western blots of in vitro translations were quantified densitometrically using ImageJ software[71]. FE was calculated as previously described, by the formula[11]

$$intensity\,(-1 - frame)/(intensity\,(-1 - frame) + intensity\,(0 - frame)). \quad (2)$$

The change in FE was calculated as a ratio of FE of each condition to the FE of no-protein control in each measurement. Experiments were repeated at least three independent times.

**Microscale thermophoresis.** Short frameshifting RNA constructs were in vitro transcribed using T7 polymerase as described above. RNAs were labeled at the 3′ end using pCp-Cy5 (Cytidine-5′-phosphate-3′-(6-aminohexyl) phosphate) (Jena Biosciences). For each binding experiment, RNA was diluted to 10 nM in Buffer A (50 mM Tris-HCl pH 7.6, 250 mM KCl, 5 mM MgCl₂, 1 mM DTT, 5% glycerol supplemented with 0.05% Tween 20 and 0.2 mg/ml yeast tRNA). A series of 16 tubes with ZAP-S dilutions were prepared in Buffer A on ice, producing ZAP-S ligand concentrations ranging from 40 pM to 2 μM. For measurements, each ligand dilution was mixed with one volume of labeled RNA, which led to a final concentration of 5.0 nM labeled RNA and 20 pM to 1 μM. The reaction was mixed by pipetting, incubated for 10 min at room temperature, followed by centrifugation at 10,000 × g for 5 min. Capillary forces were used to load the samples into Monolith NT.115 Premium Capillaries (NanoTemper Technologies). Measurements were performed using a Monolith Pico instrument (NanoTemper Technologies) at an ambient temperature of 25 °C. Instrument parameters were adjusted to 5% LED power, medium MST power, and MST on-time of 2.5 s. An initial fluorescence scan was performed across the capillaries to determine the sample quality and afterward, 16 subsequent thermophoresis measurements were performed. Data of three independently pipetted measurements were analyzed for the ΔFnorm values determined by the MO. Affinity Analysis software (NanoTemper Technologies). Graphs were plotted and binding affinities were calculated using GraphPad Prism 9.2.0 software.

**Electrophoretic mobility shift assay (EMSA).** EMSAs to visualize the stoichiometry of ZAP-S binding to SARS-CoV-2 PRF RNA variants were performed as described previously with some modifications[48]. Briefly, 100 nM RNA labelled with Cy5 at the 3′end was incubated with serial dilutions of ZAP-S in Buffer A supplemented with 5% glycerol. Reactions were separated by 0.5% agarose electrophoresis in 1× TBE prior to visualization using a Typhoon7000 imager (GE Healthcare).

**Dimethyl sulfate mutational profiling with sequencing (DMS-MaPseq)**. 50 ng of RNA was first heat denatured at 90 °C for 2 mins followed by chilling on ice for 2 min. RNA was then refolded in 50 mM HEPES pH 7.6, 250 mM KCl, 0.05 mM EDTA, 5% glycerol, 5 mM MgCl$_2$, 0.2 mg/ml yeast tRNA, and 20 U of RNasin for 15 min at 37 °C. Recombinant ZAP-S was buffer exchanged into 250 mM KCl, 50 mM HEPES/KOH pH 7.6, 0.05 mM EDTA, 5% glycerol and added to a final concentration of 5 μM and incubated at 25 °C for 10 min. DMS was diluted in EtOH to a working concentration of 1.7 M. 1/10 volume of DMS working stock was added to the samples to make a final concentration of 170 mM in a total volume of 30 μl. Samples were incubated at 37 °C for 6 min and then quenched with 30 μl of beta-mercaptoethanol (from a 14.2 M stock). For the untreated control, EtOH was used instead of DMS. RNA was then purified by Trizol LS according to the manufacturer's instructions.

Probed and control RNA was reverse transcribed using 40 U MarathonRT in RT Buffer (50 mM Tris–HCl pH 8.3, 200 mM KCl, 5 mM DTT, 20% glycerol, 1 mM MnCl$_2$), 0.5 mM dNTP mix, 2 μM primer [GGcgaagagcaggttgcaggat] and 8 U of RNasin in a final volume of 25 μM. Reverse transcriptions were carried out at 42 °C for 3 h. cDNA was diluted 1/10 with nuclease free water and PCR amplified using PrimeSTAR GXL DNA polymerase. Reaction conditions were 8 μl of diluted cDNA, 1× GXL reaction buffer, 0.2 mM dNTPs, 0.25 μM of forward [TCGTCGGCAGCGTCAGcttcgcaggagctcgacagctac] and reverse primer [GTCTCGTGGGCTCGGAGGGcgaagagcaggttgcaggat], 0.025 U/μl of polymerase in a final volume of 25 μl. Cycling conditions were 30 sec at 98 °C then 25 cycles of 10 s at 98 °C, 15 s at 60 °C and 15 s at 68 °C then 68 °C for 5 min. PCR products were verified on a 1.5% agarose gel followed by column purification (NucleoSpin Gel and PCR Clean-up, Macherey-Nagel) according to the manufacturer's instructions. A final indexing PCR was carried out using 40 ng of PCR products using Illumina Nextera DNA CD indexes (96 Indexes, 96 Samples, Illumina). Reaction conditions were 40 ng of purified PCR product, 1× Q5 reaction buffer, 0.2 mM dNTPs, 2.5 μl of indexing primer, 0.02 U/μl of Q5 polymerase in a final volume of 15 μl. Cycling conditions were 30 s at 98 °C then 5 cycles of 10 s at 98 °C, 15 s at 60 °C and 15 s at 68 °C then 68 °C for 5 min. Indexed PCR products were verified on a 1.5% agarose gel, pooled together in an equimolar ratio, before final purification on a 1.5% agarose gel. Pooled indexed sequencing library was quantified using the NEBNext library Quant Kit for Illumina and sequenced on an Illumina Miniseq using a 150 cycle High Output reagent kit.

DMS-MaP-seq data was trimmed using cutadapt[72] and aligned to the reference sequence using bowtie2[73]. Cutadapt parameters were "–nextseq-trim 20 –max-n 0 -a atcctgcaacctgctcttcgcc -A gtagctgtcgagctcctgcgaag". Bowtie2 parameters were "-D 20 -R 3 -N 1 -L 15 -i S,1,0.50 --rdg 5,1 --rfg 5,1 --maxins 600". Further analysis was carried out using the rf-count and rf-norm modules of RNA Framework package[74]. rf-count parameters were "-po -pp -m -ds 75 -q 30 -es -cc". rf-norm parameters were "-rb AC -sm 3 -nm 1". Data were plotted onto PK structures using StructureEditor (version 1.0).

**Microscopy**. HEK293 cells were cultured on glass slides and transfected as described above. The cells were fixed with 4% paraformaldehyde in 1× PBS for 15 min at room temperature. After washing with 1X PBS, cells were mounted in ProLong Antifade Diamond without DAPI (Invitrogen). Microscopy was performed using a Thunder Imaging System (Leica) using 40% LED power and the 40X objective. EGFP was excited at 460–500 nm and detected at 512–542 nm. mCherry was excited at 540–580 nm and detected at 592–668 nm. The images were processed with the LasX software (Leica). For immunofluorescence, Huh-7 cells naïve or overexpressing ZAP-S were pre-stimulated or infected as mentioned above. Cells were fixed with 6% paraformaldehyde in PBS for 1 h at room temperature, followed by washing with PBS. Cells were permeabilized with 0.1% Triton X-100 in PBS for 10 min at room temperature, washed with PBS, and blocked with 2% BSA in PBS for 1 h. Antibody labelling was performed with recombinant anti-nucleocapsid protein SARS-CoV-2 (Abcalis, Germany; #ABK84-E2-M) and secondary antibody anti-mouse Alexa488 (Cell Signaling Technology, USA; #4408), each step was followed by three washing steps with PBS containing 0.05% Tween-20. Finally, cells were overlaid with Vectashield Mounting Medium (Biozol, Germany).

**Polysome profiling analysis**. A plasmid expressing ZAP-S N-terminally tagged with a His-tag was transfected into HEK293 cells using PEI, as described above. To check endogenous ZAP-S expression, HEK cells were transfected with a plasmid containing the same backbone and His-tag. At 24 h post-transfection, cycloheximide (VWR) was added to the medium at a final concentration of 100 μg/ml to stop translation. Approximately 107 HEK cells were lysed with 500 μl lysis buffer (20 mM Tris-HCl pH 7.4, 150 mM NaCl, 5 mM MgCl$_2$, 1 mM DTT, 100 μg/ml Cycloheximide, 1% Triton X-100), and the lysate was clarified by centrifugation at 170,000 × g for 10 min at 4 °C. Polysome buffer (20 mM Tris-HCl pH 7.4, 150 mM NaCl, 5 mM MgCl$_2$, 1 mM DTT, 100 μg/ml Cycloheximide) was used to prepare all sucrose solutions. Sucrose density gradients (5–45% w/v) were freshly made in SW41 ultracentrifuge tubes (Beckman) using a Gradient Master (BioComp Instruments) according to manufacturer's instructions. The lysate was then applied to a 5–45% sucrose continuous gradient and centrifuged at 35,000 rpm (Beckman Coulter Optima XPN) for 3 h, at 4 °C. The absorbance at 254 nm was monitored and recorded and 500 μl fractions were collected using a gradient collector

(BioComp instruments). The protein in each fraction was pelleted with trichloroacetic acid, washed with acetone, and subjected to western blotting, as described above. For polysome profiling analysis of RRL a similar procedure was followed except SARS-CoV-2 mRNA was in vitro transcribed and translated in RRL as described above for 20 min at 30 °C and 300 μl of this lysate was applied to a sucrose gradient.

**Ribosome pelleting assay**. Calu-3 lysates were prepared as described above. 300 μl of the lysate was loaded onto a 900 μl 1 M sucrose cushion in polysome buffer (described above) in Beckman centrifugation tubes. Ribosomes were pelleted by centrifugation at 75,000 rpm for 2 h, at 4 °C, using a Beckman MLA-130 rotor (Beckman Coulter Optima MAX-XP). After removing the supernatant, ribosome pellets were resuspended in polysome buffer and were used for western blotting, as described above.

**Optical tweezers constructs**. 5′ and 3′ DNA handles, and the template for in vitro transcription of the SARS-CoV-2 putative PK RNA were generated by PCR using the pMZ_lambda_OT vector. The 3′ handle was labeled during the PCR using a 5′ digoxigenin-labeled reverse primer. The 5′ handle was labeled with Biotin-16-dUTP at the 3′ end following PCR using T4 DNA polymerase. The RNA was in vitro transcribed using T7 RNA polymerase. Next, DNA handles (5′ and 3′) and in vitro transcribed RNA were annealed in a mass ratio 1:1:1 (5 μg each) by incubation at 95 °C for 10 min, 62 °C for 2 h, 52 °C for 2 h and slow cooling to 4 °C in annealing buffer (80% formamide, 400 mM NaCl, 40 mM HEPES, pH 7.5, and 1 mM EDTA, pH 8) to yield the optical tweezer suitable construct (Fig. 4E). Following the annealing, samples were concentrated by ethanol precipitation, pellets were resuspended in 40 μl RNase-free water, and 4 μl aliquots were stored at −80 °C until use.

**Optical tweezers data collection and analysis**. Optical tweezers (OT) measurements were performed using a commercial dual-trap platform coupled with a microfluidics system (C-trap, Lumicks). For the experiments, optical tweezers (OT) constructs were mixed with 4 μl of polystyrene beads coated with antibodies against digoxigenin (AD beads, 0.1% v/v suspension, Ø 2.12 μm, Spherotech), 10 μl of assay buffer (20 mM HEPES, pH 7.6, 300 mM KCl, 5 mM MgCl$_2$, 5 mM DTT and 0.05% Tween 20) and 1 μl of RNase inhibitor. The mixture was incubated for 20 min at room temperature in a final volume of 19 μl and subsequently diluted by the addition of 0.5 ml assay buffer. Separately, 0.8 μl of SA-coated polystyrene beads (SA beads, 1% v/v suspension, Ø 1.76 μm, Spherotech) were mixed with 1 ml of assay buffer. The flow cell was washed with the assay buffer, and suspensions of both SA beads and the complex of OT construct with anti-digoxigenin (AD) beads were introduced into the flow cell. During the experiment, an AD bead and a SA bead were trapped and brought into proximity to allow the formation of a tether. The beads were moved apart (unfolding) and back together (refolding) at a constant speed (0.05 μm/s) to yield the force-distance (FD) curves. The stiffness was maintained at 0.31 and 0.24 pN/nm for trap 1 (AD bead) and trap 2 (SA bead), respectively. For experiments with ZAP-S protein, recombinantly expressed ZAP-S was diluted to 400 nM in assay buffer and introduced to the flow cell. FD data were recorded at a rate of 78125 Hz.

Raw data files were processed using our custom-written python algorithm called Practical OT Analysis TOol. In brief, were first down sampled by a factor of 20 to speed up subsequent processing, and the noise was filtered using Butterworth filter (0.05 filtering frequency, filter order 2). FD curves were fitted using a custom written Python script, which is based on Pylake package provided by Lumicks (https://lumicks-pylake.readthedocs.io/). For data fitting, we employed a combination of two worm-like chain models (WLC1 for the fully folded double-stranded parts and WLC2 for the unfolded single-stranded parts) as described previously[50]. Firstly, the initial contour length of the folded RNA was set to 1256 ± 5 nm, and the persistence length of the double-stranded part was fitted[50]. Then, the persistence length of the unfolded RNA was set to 1 nm, and the contour length of the single-stranded part was fitted. The work performed on the structure while unfolding or refolding was calculated as difference between area under curve (AUC) of the fit for folded region and AUC of the fit for unfolded region, counted from the beginning of the FD curve till the unfolding step coordinates. To be able to compare the effect of protein presence on different structures we decided to normalize the refolding work in each pair (protein−/protein+) to the protein-sample. We used the PK+ IMP3 value as molecular crowding control and further normalized all the ZAP+ values to it. This allowed us to quantitatively compare the effect of ZAP on different RNA molecules. Data were statistically analyzed, and the results were plotted using Prism 9.2.0 (GraphPad).

**qRT-PCR**. Total RNA was isolated as described previously[75], and the reverse transcription using RevertAid (Invitrogen) was primed by oligo(dT). Reactions of quantitative real-time PCR (qRT-PCR) were set-up using POWER SYBR green Master-mix (Invitrogen) according to manufacturer's instructions and analyzed on the CFX96 Touch Real-Time PCR Detection System (Bio-Rad) under the following cycling condition: 50 °C for 2 min, 95 °C for 2 min, followed by 40 cycles of 95 °C for 15 s and 60 °C for 30 s, and ending with a melt profile analysis. The fold change in mRNA expression was determined using the $2^{-\Delta\Delta C_t}$ method relative to the values in uninfected samples, after normalization to the housekeeping gene

(geometric mean) GAPDH. Statistical analysis was conducted comparing ΔC$_t$ values of the respective RNA in uninfected and infected cells and results were plotted using Prism 9.2.0 (GraphPad).

**Quantification and statistical analysis**. All statistical analysis and software used have been mentioned in the Figure Legends and Materials & Methods. Ordinary one-sided ANOVA was followed by a Brown-Forsythe test to ensure equal variance among the samples. Finally, a Dunnett's multiple comparisons test was employed to identify the differentially regulated conditions compared to our control constructs. Statistical analysis was performed using GraphPad Prism version 9.2.0. Measurements from the in vitro western blot assay and in vivo dual fluorescence assay resulted from three technical replicates. Measurements from single-molecule experiments resulted from a specified number (n) of traces from a single experiment. For the ensemble MST analysis, all analysis for ΔFnorm from three individual replicates was performed in Nanotemper MO. Affinity software. Data were plotted and KD was determined using Graphpad Prism version 9.2.0 nonlinear regression, binding-saturation function.

**Reporting summary**. Further information on research design is available in the Nature Research Reporting Summary linked to this article.

## Data availability
The data that support this study are available from the corresponding author upon reasonable request. Additional data associated with force spectroscopy and DMS-Seq are deposited in Mendeley Data: https://doi.org/10.17632/c7rbxb86k2.1. The mass spectrometry proteomics data have been deposited to the ProteomeXchange Consortium via the PRIDE partner repository[76] with the dataset identifier PXD029656. Source data are provided with this paper.

## Code availability
Custom scripts were employed to process optical tweezers data. Python algorithm called Practical Optical Tweezers Analysis TOol is available on Github (POTATO, https://github.com/lpekarek/POTATO.git,)[77].

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

## Acknowledgements

U.R. and L.P. contributed equally to this work. We thank Dr. Zeljka Macak-Safranko and Prof. Alemka Markotic (University of Zagreb) for providing the SARS-CoV-2 virus isolate prior to publication. We thank Dr. Andreas Schlundt for kind gifts of IGF2BP3 and SUMO proteins (Goethe University, Frankfurt, Germany). We thank Dr. Joop van den Heuvel (HZI) for his helpful suggestions on protein purification. We thank Dr. Anke Sparmann, Prof. Jörg Vogel, Prof. Lars Dölken, Prof. Utz Fischer and Prof. Thomas Pietschmann for critical reading of the manuscript. We thank expert technical assistance by Tatyana Koch (HIRI-HZI). We thank Ayse Barut for cell maintenance for infection studies (HZI). We thank Dr. Andreas Schlosser and Stephanie Lamer from the Rudolf Virchow Center for the LC-MS/MS analysis. Figures were partially generated using BioRender.com (licensed for commercial printing to A.K.). This project is funded fully or in part by the Helmholtz Association. L.C.S. funded through MWK Niedersachsen Grant Nr. 14-76103-184 CORONA-2/20. N.C. received funding from the European Research Council (ERC) Grant Nr. 948636.

## Author contributions

M.Z. and A.K. designed and cloned the constructs, purified proteins, and performed most of the biochemical experiments. L.P., S.B. and N.C. designed the OT constructs, L.P. performed most of the single molecule experiments and processed the data with the help of S.B. S.B. and L.P. have written the scripts for automatized analysis of the single molecule data. U.R. performed the SARS-CoV-2 infection assays and collected lysates for downstream biochemical analysis. L.Y. and R.S. performed DMS-MaPseq experiments and analyzed the data. L.C.-S., R.S., and N.C. supervised the study. M.Z., A.K., L.P. and N.C. wrote the paper. All authors contributed to the review and editing of the final paper.

## Funding

## Competing interests

The authors declare no competing interests.
