## [Peer Review File · Nature Communications]

The short isoform of the host antiviral protein ZAP acts as an inhibitor of SARS-CoV-2 programmed ribosomal frameshiftingREVIEWER COMMENTS

Reviewer #1 (Remarks to the Author):

Zimmer et al. describe the discovery and characterization of cellular proteins that bind the programmed -1 ribosomal frameshift (-1 PRF) signal derived from the SARS-CoV-2 virus. Starting with an RNA affinity pulldown experiment, they identify a number of host proteins previously known to interact with RNA, including some that have been previously shown to be important for viral translation and activation of the innate immune response. This was followed up with assays to monitor the effects of these proteins on SARS-CoV-2 mediated -1 PRF, further whittling down the number of candidates. After another round of selection based on changes in expression in virus infected cells and -1 PRF inhibitory activity, they focused on ZAP-S, which is known to target interferon mRNAs for degradation. Surprisingly, ZAP-S appeared to specifically inhibit -1 PRF mediated by SARS-CoV-2 and the nearly identical SARS-CoV elements: it did not affect -1 PRF promoted by two other coronaviruses (MERS-CoV and Bat-CoV273), nor of -1 PRF signals of other origin. Followup experiments showed that ZAP-S can inhibit SARS-CoV-2 -1 PRF in vitro, that it directly interacts with the -1 PRF motif, and a series of biophysical experiments were used to characterize its binding affinity and ability to prevent re-folding of the SARS-CoV-2 -1 PRF stimulating mRNA pseudoknot. Additionally, pulldown experiments show that it interacts with the ribosome and identified additional interacting proteins. A final series of experiments demonstrated that ZAP-S can promote an approximately 10-fold inhibition of SARS-CoV-2 replication in Huh7 cells. In general, this is an outstanding piece of research.

Major Comments.

1. This work has great breadth but not such great depth. It is not clear whether it is about ZAP-S or about frameshifting. In particular, the final section detailing what pulls down with ZAP-S and its association with the ribosome seem to be peripheral and distracting to what (from the title) should be a story about this cellular anti-frameshifting story. There seems to be two different narratives competing here. The authors should consider separating them into two distinct papers.
2. Lines 286-7 and Fig. 4B: The binding of ZAP-S to a mutant of the bulged A was not reported. Why is this mutant mentioned here? Indeed, the only mutants that were assayed were those that fully disrupted the structure. This is an example of lack of depth. The authors should assay a few well-selected ZAP-S mutants, e.g. pick from a few that are described in the recent Science paper from the Ban group, in order to obtain a clearer picture of the structural requirements for ZAP-S binding to the SARS-CoV-2 pseudoknot. These mutants should also be assayed using the optical tweezers (seen next comment), and microscale thermophoresis.
3. Fig. 4E. The unfolding and refolding profiles of the pseudoknot that are shown here do not compare well with the profiles generated by the Woodside lab. They have a "quick and dirty" look to them. Likewise Figs. S4A and B. In particular, the two step unwinding profile is completely absent in the wild-type, and is barely apparent in the refolding profile. These data are unconvincing. This is important because it is the most convincing data for specific binding of ZAP-S to this element and is the basis for the model shown in Fig. 6 of ZAP-S altering the pseudoknot structure. It would be nice to see cleaner data and/or an orthogonal approach.
4. The specificity of ZAP-S for the SARS-CoV family of -1 PRF signals is surprising given that prior to 2003, humans had never been exposed to these viruses. Evolution does not tend to be "anticipatory". Therefore, it is logical to hypothesize that this activity of ZAP-S evolved in response to another viral -1 PRF signal. There are four other, well-established human coronaviruses which cause approximately 30% of common colds: 229E (alpha coronavirus); NL63 (alpha coronavirus); OC43 (beta coronavirus); and HKU1 (beta coronavirus). A reasonable hypothesis is that this function of ZAP-S may have evolved in response to one or a few of them. This should be tested.

Minor comments:

1. The term 'for the first time' is considered to be 'bad form' in scientific writing. Please try not to use it.
2. Lines 233-235: the statement that even a 10% change in -1 frameshifting inhibited SARS-CoV viral propagation and reduced infectivity is incorrect. In the cited reference, the mutant with the smallest change in frameshifting that was tested with regard to virus propagation conferred an ~65% reduction,

not 10%.

5. A 10-fold reduction in viral titers is not considered to be “big” by virology standards. 3-orders of magnitude or greater are considered to be the threshold for a strong effect on virus replication.

6. Line 462: the data show a significant amount of residual frameshifting, indicating that the roadblock effect is lessened, not abrogated.

Reviewer #2 (Remarks to the Author):

The manuscript by Zimmer et al. reports the discovery of host proteins (including ZAP-S) that bind to SARS-CoV-2 viral RNA programmed ribosomal frameshifting stimulating RNA structure through pull-down assays. In this work, the authors focused on characterizing ZAP-S by cell culture reporter gene assays, viral replication and cell-free translation assay and ensemble binding studies. The data suggest that ZAP-S interacts with frameshifting stimulating viral RNA pseudoknot in SARS-CoV-2 infected cells, and inhibits frameshifting and viral replication. In addition, the authors utilized single-molecule optical tweezers techniques to study how affects RNA pseudoknot folding. The authors need to make the below clarifications and corrections before the paper may be published.

Page 2, line 57, A related ref *Biochemistry* 2018, 57, 149–159 provides insights into how dsRNA binders may stimulate ribosomal frameshifting.

Page 6, line 156, Fig. 2D may not be referred to here.

Page 6, line 159, the exact sequences and structures of the mutants of the RNAs should be shown in SI.

Page 7, line 172, The authors may have a discussion about why ZAP mRNA transcription is activated.

Figure 2C, The authors may discuss how ZAP protein overexpression affects the mRNA levels of the report genes as well as viral RNA expression levels. The reason is that ZAP is known to be involved in modulating RNA stability (page 7, line 176). The mRNA levels may affect ribosome density and thus frameshifting efficiency as shown before. In addition, repression of translation initiation by ZAP-S (as stated in page 17, line 443) may also affect frameshifting efficiency.

Figure 3B, It seems that ZAP-S inhibits the production of zero-frame protein expression. Again, the authors may check if ZAP-S affects mRNA integrity here.

Figure 4, In the single-molecule experiments, a control RNA, e.g., a stem-loop or a different pseudoknot may be added. It seems that ZAP-s affects the folding of the stem-loop within the SARS-CoV-2 pseudoknot structure. Thus, it is very likely that a non-specific molecular crowding effect causes the slowing down of the folding of stem-loops within the pseudoknot. Such potential effect was discussed in ref: *JACS* 2018, 140, 8172-8184.

The statement in page 17, line 435 about “structural plasticity” is inconsistent with the model proposed in page 18, second paragraph. In the “structural plasticity” model, dynamics of multiple secondary structures are proposed to be important in enhancing ribosomal frameshifting. In fact, a recent *Science* paper (DOI: 10.1126/science.abf3546) clearly shows that specific mRNA pseudoknot-ribosome interaction is critical for stimulating ribosomal frameshifting. In addition, the frameshifting assay data reported in this *Science* paper suggest that a small molecule did not directly affect frameshifting, although the small molecule was previously reported to be a frameshifting inhibitor and affect single-molecule mechanical folding of pseudoknots.

Reviewer #3 (Remarks to the Author):

The authors identify the short isoform of ZAP as a binder to SARS-CoV2 frame shift element and show that the interaction influences RNA folding. Furthermore, they show that overexpression of ZAP results in changing frameshift frequency and ZAP associates with the ribosome.

Major

Figure 2D: Why has the RNA levels only be determined for ZAP and not by isoform specific primers? In principle, the amount of information on mRNA changes upon infection should not be just 3 transcripts, but the authors should conduct an RNA-Seq experiment. It is just simple enough nowadays and can be even outsourced. It would give a much better picture on the results of infection of their cell line and also be useable to check for upregulated proteins among all their interactors. The ANOVA tests need to be followed by e.g. a Tukey's Posthoc test to actually judge which are the differentially regulated conditions. Applies to several occasions throughout the manuscript.

Figure 2C: How is controlled that the amount of overexpressed protein is correlated to the effect size? The authors place such an emphasis on PRF efficiency to select their candidates that this needs to be controlled. Reading through the manuscript, it is not clear to me why ZAP-S should have a stronger effect than ZAP-L, but one explanation might be actual expression levels in this assay here. It would have been a much stronger study if ZAP-L would have been taken along and for a publication at this level, I would like to see the data also for ZAP-L.

For me, it looks like a little overselling of the short ZAP isoform, while there is no proof that the long ZAP isoform should not work either. The authors should acknowledge and discuss this better in the manuscript. (Especially the comment that "ZAP-S is specific" I don't see justified with the presented data.)

I also would remind the authors to not overinterpret that only CoV and ZAP worked in their assay. While they tested a few structures, they can't completely rule out that there are more. It is a kind of lame argument on my side, but I don't think that the human ZAP-S isoform just evolved to recognize CoV and CoV2.

For Figure 3, the authors should also include data from IGF2BP1 as they seem to suggest that it has a functional relevance and not just IGF2BP3. Why weren't they actually tested in Figure 2C, given they are very similar, but show different effects?

Figure 4: For this assay (E and F) IGF2B3 should be used as a control to rule out just general effects of protein in the lysate. The authors already expressed IGF2BP3, thus it is not a large effort to include a more appropriate control.

Why not endogenous studies as a ZC3HAV1 antibody (Fig. S2D) was available? This limits in my view the value of the study. There could be overexpression artefacts. A orthogonal knock-down/knock-out analysis is also not available.

The association with ribosomes has only been done biochemically and looks like a case of sticky behavior. Can this be validated with microscopy or alternatively better suited assay? For sure another RNA-binding protein as negative control is missing in Figure 5b and 5c experiments (actin is not enough).

Overall: I think that the inclusion of RNA-Seq, adding appropriate controls to experiments in Figure 3, Figure 4 and Figure 5 plus the data on ZAP-L is crucial and should be a prerequisite to accepting this study.

Minor

I would suggest to rename the heading: "Revealing the short isoform of the host antiviral protein ZAP as an ..." to avoid the misleading information that ZAP-S is a full protein name.

Abstract: Also in the text, I would add: "reveal that the short isoform of zinc-finger antiviral protein (ZAP-S) is" (afterwards the use of ZAP-S is justified as introduced as an isoform definition here).

Abstract: It is unclear to me why it is a "de novo host encoded factor" and not just a "host encoded factor".

In Figure 1B, the SSB writing is not clear enough visible.

The single-molecule argument might be misleading (see e.g ZAP interacts with translating ribosome, line 2) as not a single molecule ZAP was studied, but a single RNA molecule.

Figure S4C: The data is not really interpretable and look more similar than the examples in S4A and

S4B. Can this be better represented? I would actually recommend to move Fig. S4D or at least the 0 part to the main figure to not have a cherry-picked representation there, but some global analysis from all measurements.

Point by Point Responses

We thank the reviewers for their time and efforts and for their thoughtful and constructive comments that allowed us to strengthen our manuscript.

The major revisions to the manuscript are as follows:

1. We have included new *in vivo* frameshifting reporter assay data on additional human coronaviruses, namely HKU1 and OC43 as well as 4 different SARS-CoV-2 mutant RNA variants. In order to strengthen the reason for not selecting ZAP-L, we provide *in vivo* data of ZAP-L effect on SARS-CoV-2 frameshift RNA variants.
2. We have performed additional single-molecule optical tweezers experiments and ensemble microscale thermophoresis analysis with these RNA variants in the presence and absence of ZAP-S and could illustrate that the Stem 2 and 3 regions of the putative pseudoknot are the main ZAP-S interaction sites. Furthermore, we have carried out structural probing experiments by DMS-MaPseq, which suggest that reactivities of these stem regions change in the presence of ZAP-S.
3. A control RNA binding protein IMP3 was tested in ensemble microscale thermophoresis and single molecule optical tweezers assays to monitor the effect of non-specific interactions, and technical artefacts on RNA. To make the point clearer that ZAP-S interferes with the energetics of the refolding of the RNA variants with different physical properties, we carried out work calculations during unfolding and refolding. We also improved the overall representation of this complex data.
4. In order to keep the storyline focused on the *in vitro* characterization of ZAP-S function on frameshifting, we de-emphasized the data showing ZAP:ribosome interactions, removed ZAP co-immunoprecipitations and present the polysome profiling data in a different context to illustrate that ZAP-S interactions with the viral RNA occur during translation *in vivo* and *in vitro*.

Point by point replies to individual reviewer comments is below. The original reviews and specific points are reproduced (in blue) for convenience.

Reviewer #1 (Remarks to the Author):

Zimmer et al. describe the discovery and characterization of cellular proteins that bind the programmed -1 ribosomal frameshift (-1 PRF) signal derived from the SARS-CoV-2 virus. Starting with an RNA affinity pulldown experiment, they identify a number of host proteins previously known to interact with RNA, including some that have previously been shown to be important for viral translation and activation of the innate immune response. This was followed up with assays to monitor the effects of these proteins on SARS-CoV-2 mediated -1 PRF, further whittling down the number of candidates. After another round of selection based on changes in expression in virus infected cells and -1 PRF inhibitory activity, they focused on ZAP-S, which is known to target interferon mRNAs for degradation. Surprisingly, ZAP-S appeared to specifically inhibit -1 PRF mediated by SARS-CoV-2 and the nearly identical SARS-CoV elements: it did not affect -1 PRF promoted by two other coronaviruses (MERS-CoV and Bat-CoV273), nor of -1 PRF signals of other origin. Follow-up experiments showed that ZAP-S can inhibit SARS-CoV-2 -1 PRF *in vitro*, that it directly interacts with the -1 PRF motif, and a series of biophysical experiments were used to characterize its binding affinity and ability to prevent re-folding of the SARS-CoV-2 -1 PRF stimulating mRNA pseudoknot. Additionally, pulldown experiments show that it interacts with the ribosome and identified additional interacting proteins. A final series of experiments demonstrated that ZAP-S can promote an approximately 10-fold inhibition of SARS-CoV-2 replication in Huh7 cells. In general, this is an outstanding piece of research.

Major Comments.

1. This work has great breadth but not such great depth. It is not clear whether it is about ZAP-S or about frameshifting. In particular, the final section detailing what pulls down with ZAP-S and its association with the ribosome seem to be peripheral and distracting to what (from the title) should be a story about this cellular anti-frameshifting story. There seems to be two different narratives competing here. The authors should consider separating them into two distinct papers.

We have now rephrased the section describing the association of ZAP-S with the ribosome and combined the polysome profiling data with *in vitro* translation experiments in Fig. 3. We hope that the implemented changes re-focus the narrative of the manuscript on the anti-frameshifting activity of ZAP-S. To this end, we also removed the co-immunoprecipitation data from the manuscript. Nevertheless, several proteins that play a role in PRF have previously been shown to interact with ribosomes (e.g., 2A, SFL). Since we see a similar association of ZAP-S with the ribosome *in vitro* and in naïve cells, we find it important to include these data here, although we agree with the reviewer that a detailed analysis of the functional relevance of ZAP-S interactions with the ribosome is beyond the scope of the current manuscript.

2. Lines 286-7 and Fig. 4B: The binding of ZAP-S to a mutant of the bulged A was not reported. Why is this mutant mentioned here? Indeed, the only mutants that were assayed were those that fully disrupted the structure. This is an example of lack of depth. The authors should assay a few well-selected ZAP-S mutants, e.g. pick from a few that are described in the recent Science paper from the Ban group, in order to obtain a clearer picture of the structural requirements for ZAP-S binding to the SARS-CoV-2 pseudoknot. These mutants should also be assayed using the optical tweezers (seen next comment), and microscale thermophoresis.

We mentioned the bulged A mainly to emphasize that Stem-loop 2 (SL2) is extremely important for frameshifting and that even a single mutation within this stimulatory RNA element can interfere with frameshifting efficiencies. Following the suggestions of the reviewer, we now expanded our work with additional mutants and performed rigorous analysis with microscale thermophoresis, DMS-MaPseq and optical tweezers.

- We took advantage of the published structure of the SARS-CoV-2 pseudoknot and generated truncation mutants of individual stem loops SL2, SL3 and SL2+3, which helped us to assess the structural requirements for ZAP-S function. While the deletion of both SL2 and SL3 completely

abolishes ZAP-S binding, deletion of the SL3 alone showed only a moderate effect on the interaction ($K_D=170$ nM). Upon deletion of the 3' distal region, which would likely prevent formation of SL2, we observed a lower binding affinity (delta SL2, $K_D=600$ nM). These data suggested two putative ZAP-S binding sites, involving the stem 2 and stem 3 regions and we confirmed these observations further by EMSAs. K_D calculations derived from two binding events (130 nM, calculated from the formula $1/K_D=1/K_{D1}+1/K_{D2}$) are within the range of our experimentally determined K_D (120 nM).

- We employed DMS-MaPseq to probe ZAP-S binding, which, in accordance with the results discussed above, also suggested altered accessibility in SL2 and SL3 regions upon ZAP-S binding (please refer to text change lines 318-325 and Suppl. Fig. 4).
- We also carried out optical tweezer experiments and analysis with the mutant RNAs in the absence and presence of ZAP-S. As discussed in detail under point #3 below, we found that unfolding of all RNA variants was not altered but the refolding of wild type was impeded by ZAP-S. Furthermore, while we could still observe this effect of ZAP-S on the refolding of delta SL2 and delta SL3 mutants, deletion of both SL2 and SL3 (delta SL2+3) completely abolished this effect of ZAP-S. (Please refer to ms text for details, lines 367 – 398, 408-413).

We carefully considered the frameshift site mutations reported by the Ban and Atkins labs. However, because we cannot fully predict the effect of these deletions on the pseudoknot fold, we felt that truncation mutants of individual pseudoknot stem loops would be the better approach to study the binding interactions of ZAP.

Altogether, these new data further define the putative ZAP-S binding sites and strengthen our original conclusion that ZAP-S binds specifically to the SARS-CoV-2 pseudoknot structure.

3. Fig. 4E. The unfolding and refolding profiles of the pseudoknot that are shown here do not compare well with the profiles generated by the Woodside lab. They have a “quick and dirty” look to them. Likewise, Figs. S4A and B. In particular, the two step unwinding profile is completely absent in the wild-type and is barely apparent in the refolding profile. These data are unconvincing. This is important because it is the most convincing data for specific binding of ZAP-S to this element and is the basis for the model shown in Fig. 6 of ZAP-S altering the pseudoknot structure. It would be nice to see cleaner data and/or an orthogonal approach.

As discussed in point #2 above, we have now included additional data using orthogonal approaches (DMS seq and MST analyses) and using new RNA variants with sequential deletions within the putative stem regions of the predicted pseudoknot (delta SL2, SL3 and SL2+3).

Further, to strengthen our single molecule optical tweezers experiments:

- We now include additional data on the wild type pseudoknot (WT PK) and the 4 additional mutant RNA variants discussed above. The total number of curves analyzed in the study has now reached >2000 (Supp. Table 5).
- We include DMS-MaPSeq as an alternative approach to probe ZAP-S binding and its effects on the overall RNA fold.
- As an additional control, we also conducted WT PK RNA experiments in the presence of an RNA-binding protein IMP3, which was identified in our proteomic screen as a weak binder. While we observe a slight effect on refolding curves with IMP3, probably due to non-specific interactions, the effect of ZAP-S on PK RNA is clearly distinct from what is seen with IMP3 and all other controls that we employed (Fig. 5I-J).
- Since some of the RNA mutants differ substantially in their unfolding-refolding pattern compared to the WT, a direct comparison of the various curves was not possible. Thus, we quantified the effect of ZAP-S on the RNA variants by calculating and comparing the ‘work done’ during folding (Fig. 5K).

- To improve the representation of the data, we now provide an overlay image of 5 different unfolding and folding curves for each RNA variant (Fig. 5D-I). We also included histograms of the change in contour length and the change in force distribution in Supplemental Fig. 5, 6.

Together with other *in vitro* ensemble measurements of RNA binding by MST and DMS-MaPseq analysis our data strongly support our initial hypothesis that an interaction with ZAP-S triggers an alteration in PK RNA refolding (refer to ms, lines 297-336; 339-434).

- Regarding potential differences between our data and the results reported by the Woodside lab (Neupane et al. 2021, PMID: 34362921), we would like to point out that there are differences in the construct design between both studies. In particular, the Woodside group used a longer 5' spacer derived from the native SARS-CoV-2 genome. Therefore, our results cannot be directly compared. In our analysis, 80% of the unfolding events were marked by a single unfolding event. In a minority of the curves, we were able to see the two-step unfolding that was reported by the Woodside lab (now included in Fig 5D, compare with Figure 2b in Neupane et al., PMID: 34362921 below). Regardless of these differences, the PK-like conformer with the contour length change value of 34.5 ± 3.0 nm was in a very good agreement with the values reported by Neupane et al. (PMID: 34362921) (35.6 ± 0.4 nm).

The authors also proposed an additional conformer, which was not prominent in our data. We speculate this state could form due to the presence of 5' spacer sequence. In that regard, Schlick et al. recently reported that the 5' region preceding Stem-loop 1 of the SARS-CoV-2 frameshifting PK can form an alternative structure by interacting with the 3' part of stem-loop 2 (PMID: 32869017). We have now included this discussion in the manuscript text (line 344-345).

Rebuttal Fig. 1 – Optical tweezers data from Neupane et al., 2021, PMID: 34362921 (**top**) and the predicted structure formed by 5' spacer (**bottom**).

4. The specificity of ZAP-S for the SARS-CoV family of -1 PRF signals is surprising given that prior to 2003, humans had never been exposed to these viruses. Evolution does not tend to be “anticipatory”. Therefore,

it is logical to hypothesize that this activity of ZAP-S evolved in response to another viral -1 PRF signal. There are four other, well-established human coronaviruses which cause approximately 30% of common colds: 229E (alpha coronavirus); NL63 (alpha coronavirus); OC43 (beta coronavirus); and HKU1 (beta coronavirus). A reasonable hypothesis is that this function of ZAP-S may have evolved in response to one or a few of them. This should be tested.

This is an interesting point. Indeed, we were also surprised that ZAP-S did not show any effect on other PRF sites we tested. As suggested by the reviewer, we have now included PRF elements from other coronaviruses – namely HKU1 and OC43. We observed a slight, albeit not statistically significant reduction in frameshift levels mediated by ZAP-S.

While coronaviral PRF elements are relatively conserved (Rebuttal Fig. 2), there are notable differences in their sequence and hence their structure. For example, SARS-CoV-1 and SARS-CoV-2 have a deletion of three nucleotides between SL1 and SL2, which might result in a different fold of the pseudoknot, as proposed by Plant et al., 2005 (PMID: 15884978). In addition, SARS-CoV-1 and SARS-CoV-2 are the only coronaviruses that share a high sequence similarity in SL3, which, according to our biophysical interaction studies and DMS-Seq results, is one of the potential ZAP-S binding sites.

Future studies in our lab are aimed at understanding the structural basis of the ZAP-S-mediated reduction of frameshift efficiency specifically in SARS-CoV-1 and -2. At this point, however, we can only speculate about the evolutionary origins of this effect. We agree with the reviewer that it is highly unlikely that this function of ZAP evolved in response to coronaviruses. ZAP is a multi-functional host protein, with several other antiviral roles, perhaps the regulation of SARS-CoV frameshifting is a serendipitous side effect.

```

SARS-CoV-2      UUUUUAAACGGGUUUGCGGUGUAAGUGCA---GCCCGUCUJACACCGUGCGGCACAGGCACUAGUACUGAUGUCGUAUACAGGGCU 83
SARS-CoV-1      UUUUUAAACGGGUUUGCGGUGUAAGUGCA---GCCCGUCUJACACCGUGCGGCACAGGCACUAGUACUGAUGUCGUCUACAGGGCU 83
MERS            UUUUUAAACGAGUCGGGGUUCUUAUUGUAAAUGCCCGAAUAGAACCUGUUCUAGUGGUUUUGUCCACUGAUGUCGUCUUUAGGGCA 86
HKU1            UUUUUAAACGGGUUCGGGGUACUAGUGUGAAUGCCCGGCUAGUACCCUGUGCUAGUGGUUUUAUCUACUGAUGUCAAUUAGGGCA 86
OC43            UUUUUAAACGGGUUCGGGGUACGAGUGUAGAUGCCCGUCUCGUACCCUGCGCCAGUGGUUUUAUCUACUGAUGUCAAUUAGGGCA 86
***** ** * ** * ** * ** * ** * ** * ** * ** * ** * ** * ** * ** * ** * ** * ** * ** * ** * **

```

Rebuttal Fig. 2 – Alignment of PRF elements of several coronaviruses. Asterisks denote perfect conservation between all sequences shown.

Minor comments:

1. The term 'for the first time' is considered to be 'bad form' in scientific writing. Please try not to use it.

Changed.

2. Lines 233-235: the statement that even a 10% change in -1 frameshifting inhibited SARS-CoV viral propagation and reduced infectivity is incorrect. In the cited reference, the mutant with the smallest change in frameshifting that was tested with regard to virus propagation conferred an ~65% reduction, not 10%.

We thank the reviewer for spotting this oversight. Corrected.

5. A 10-fold reduction in viral titers is not considered to be “big” by virology standards. 3-orders of magnitude or greater are considered to be the threshold for a strong effect on virus replication.

While we agree that the effect of ZAP overexpression on viral titers (~20-fold) might be considered modest, it is nevertheless reproducible and statistically significant. It is important to note that ZAP is only one of many other interferon stimulated genes (ISGs) that are induced upon infection as part of the host defense system, and therefore its overall contribution might be small, but functionally relevant. In order to strengthen our conclusions that ZAP-S impact viral replication, we have now also carried out an immunofluorescence assay

to detect the viral N protein, which is an early marker of SARS-CoV-2 infection. ZAP-S decreases the expression of N, further indicating that virus replication is impaired.

6. Line 462: the data show a significant amount of residual frameshifting, indicating that the roadblock effect is lessened, not abrogated.

Changed.

Reviewer #2 (Remarks to the Author):

The manuscript by Zimmer et al. reports the discovery of host proteins (including ZAP-S) that bind to SARS-CoV-2 viral RNA programmed ribosomal frameshifting stimulating RNA structure through pull-down assays. In this work, the authors focused on characterizing ZAP-S by cell culture reporter gene assays, viral replication and cell-free translation assay and ensemble binding studies. The data suggest that ZAP-S interacts with frameshifting stimulating viral RNA pseudoknot in SARS-CoV-2 infected cells, and inhibits frameshifting and viral replication. In addition, the authors utilized single-molecule optical tweezers techniques to study how affects RNA pseudoknot folding. The authors need to make the below clarifications and corrections before the paper may be published.

Page 2, line 57, A related ref Biochemistry 2018, 57, 149–159 provides insights into how dsRNA binders may stimulate ribosomal frameshifting.

We thank the reviewer for this comment. We added a brief discussion on this topic and included the reference (lines 53 – 55).

Page 6, line 156, Fig. 2D may not be referred to here.

Corrected.

Page 6, line 159, the exact sequences and structures of the mutants of the RNAs should be shown in SI.

Schematics of the RNA variants used in this study have been included in Fig. 4A, B and the exact sequences of the RNA variants are provided in Supplementary Table 3.

Page 7, line 172, The authors may have a discussion about why ZAP mRNA transcription is activated.

ZAP-S is an interferon-induced gene that is part of the innate immune response. The transcriptional regulation of ZAP has now been discussed in page 6 of the revised manuscript.

Figure 2C, The authors may discuss how ZAP protein overexpression affects the mRNA levels of the report genes as well as viral RNA expression levels. The reason is that ZAP is known to be involved in modulating RNA stability (page 7, line 176). The mRNA levels may affect ribosome density and thus frameshifting efficiency as shown before. In addition, repression of translation initiation by ZAP-S (as stated in page 17, line 443) may also affect frameshifting efficiency.

In the original manuscript, we had demonstrated that ZAP-S overexpression does not affect the stability of the reporter mRNA in cells (Supplementary Fig. 2B). To further test whether ZAP-S influences translation or ribosome density in cells, we compared polysome profiles in control and ZAP-S overexpressing cells but observed no difference (Rebuttal. Fig 3). We also checked whether the amount of 80S/polysomes change in reticulocyte lysates translating the SARS-CoV-2 reporter mRNA in the presence or absence of ZAP-S (Fig 3D). Again, we observed very similar polysome profiles. Combined, these data indicate that the effect of ZAP-S on frameshifting is not due to degradation of the reporter mRNA, a global decrease in the rate of translation or ribosome density (Supplementary Fig. 2B, please refer to text change in lines 262- 267).

As pointed out by the reviewer, it has been reported that ZAP-S might bind to 3' UTR of mRNAs and recruit RNA surveillance factors to regulate viral and some host RNAs. How and whether ZAP-S affects global translation or RNA surveillance during coronavirus infections is an interesting question, which we are actively pursuing in collaboration with other groups, but this analysis goes beyond the scope of the current

manuscript, which is focused on the effects of ZAP-S on coronaviral frameshifting. In that context, we did not observe an effect on RNA stability of reporter constructs in our experimental set-up.

Rebuttal Fig. 3 Polysome profiles of naïve HEK cells vs HEK cells overexpressing ZAP-S.

Figure 3B, It seems that ZAP-S inhibits the production of zero-frame protein expression. Again, the authors may check if ZAP-S affects mRNA integrity here.

As mentioned above, ZAP-S overexpression did not affect the abundance of the reporter mRNA in cells (Supplementary Fig. 2B). In addition, we did not observe a difference in the expression levels of GFP (0-frame) and mCherry (0-frame) control constructs in the presence and absence of ZAP-S.

The decrease in the 0 frame controls seen *in vitro* is likely an artifact of the *in vitro* translation systems when high concentrations of ZAP-S are used. Other studies have previously observed a similar effect upon increasing concentrations of *trans*-acting proteins 2A and SFL in *in vitro* reporter assays (Naphine et al 2019, 2021; PMID: 28593994, PMID: 34202160) (Rebuttal Fig. 4). To account for this effect, frameshifting efficiencies are calculated by dividing the intensity of the –1 frame product by the sum of the intensity of the 0 and –1 frame product. Therefore, the calculated frameshifting efficiency is normalized to the 0-frame band and therefore independent of any effect on the mRNA integrity (please refer to Methods).

Rebuttal Fig. 4 Representative *in vitro* translation assays with increasing concentrations of *trans*-regulatory proteins. Upper panel: TMEV 2A protein (from Naphine et al., 2019), lower panel: Shiftless protein (from Naphine et al., 2021).

Figure 4, In the single-molecule experiments, a control RNA, e.g., a stem-loop or a different pseudoknot may be added. It seems that ZAP-s affects the folding of the stem-loop within the SARS-CoV-2 pseudoknot structure. Thus, it is very likely that a non-specific molecular crowding effect causes the slowing down of the folding of stem-loops within the pseudoknot. Such potential effect was discussed in ref: JACS 2018, 140, 8172-8184.

We followed the reviewer's suggestion and have now included 4 additional RNA mutants of the putative SARS-CoV-2 PK to dissect the interactions between ZAP-S and the SARS-CoV-2 RNA. As an example of an additional stem-loop control, we employed the Δ SL2+SL3 mutant, which is predicted to fold into SL1 (about 30 nt). ZAP-S did not substantially affect refolding of this stem loop alone. We also employed a PK variant, referred to as 'compensatory mut.', which is predicted to form a stack of 4Gs in the stems. Notably, the unfolding of RNA occurred at higher forces suggesting a stabilization of the RNA fold. This variant still interacts with ZAP-S with similar affinity as the wt PK, but we did not observe a drastic change in refolding which suggest either the structure or the local thermodynamic stability of the RNA is different. An alternative could also be that the faster kinetics of RNA refolding in this pseudoknot masks the effect of ZAP-S (please also see lines 465-467).

To rule out the possibility that the energetics of refolding are altered simply due to molecular crowding, we conducted measurements of the PK RNA in the presence of IMP3, another unrelated RBP. IMP3 was identified as a weak binder in our proteomic screen and MST confirmed its lower affinity compared to ZAP (K_D around 800 nM, Supplementary Fig. 3H, also ms text lines 305-318). In this case, we observed a very mild effect on RNA refolding, which was likely due to non-specific steric interactions or molecular crowding. Similar effects were also observed with the RNA variants Δ SL2+SL3 in the presence of ZAP-S, although ZAP-S clearly did not bind to that RNA variant or affect frameshifting (Supplementary Fig. 4F, ms lines 399-413).

To quantify these effects, we now calculate the work during refolding in the presence of ZAP-S (Fig. 5K). The normalized work values of the PK and mutant RNAs in the presence of control protein IMP3 were very similar to the values obtained for the Δ SL2+3 mutant and the compensatory mutant in presence of ZAP-S (Fig. 5K). Based on these data, we conclude that the effect of ZAP-S on PK RNA refolding is distinct and stronger than observed for the various control conditions, hence molecular crowding or non-specific steric contributions are unlikely to account for the observed effect.

The statement in page 17, line 435 about "structural plasticity" is inconsistent with the model proposed in page 18, second paragraph. In the "structural plasticity" model, dynamics of multiple secondary structures are proposed to be important in enhancing ribosomal frameshifting. In fact, a recent Science paper (DOI: 10.1126/science.abf3546) clearly shows that specific mRNA pseudoknot-ribosome interaction is critical for stimulating ribosomal frameshifting. In addition, the frameshifting assay data reported in this Science paper suggest that a small molecule did not directly affect frameshifting, although the small molecule was previously reported to be a frameshifting inhibitor and affect single-molecule mechanical folding of pseudoknots.

We revised our discussion of the model accordingly and further discuss the possible effects of *trans* acting factors (page 20). Indeed, not every protein that we identified in our proteomic screen as an interactor with the frameshift PK also altered frameshifting levels (these were previously presented in SI now added in Fig. 2C). In addition, even if we detect binding of ZAP-S to some of the RNA variants tested, we did not necessarily see a difference in the refolding of the RNA (comp. mut., Fig 5G). Therefore, binding to the RNA alone is not sufficient for altering RNA dynamics or frameshifting. We are currently pursuing structural studies to address how the structural dynamics of the RNA and *trans*-acting factors modulate translation, but resolving this issue is clearly beyond the scope of the current work.

Reviewer #3 (Remarks to the Author):

The authors identify the short isoform of ZAP as a binder to SARS-CoV2 frame shift element and show that the interaction influences RNA folding. Furthermore, they show that overexpression of ZAP results in changing frameshift frequency and ZAP associates with the ribosome.

Figure 2D: Why has the RNA levels only be determined for ZAP and not by isoform specific primers? In principle, the amount of information on mRNA changes upon infection should not be just 3 transcripts, but the authors should conduct an RNA-Seq experiment. It is just simple enough nowadays and can be even outsourced. It would give a much better picture on the results of infection of their cell line and also be useable to check for upregulated proteins among all their interactors.

To address this point, we took advantage of published RNAseq data and compared gene expression levels in infected Huh7.5.1, Calu-3 cells and patient samples (Blanco-Melo et al., 2020; Sun et al., 2021 as referenced in the ms). These data allowed us to better illustrate differences in expression levels of the core interactors identified in our screen. ZAP was still the most enriched transcript in the RNAseq data sets, which aligned well with our qRT-PCR analysis (Supplementary Fig. 1E and F) (please refer to lines 170- 181).

We also looked at the Blanco-Melo data to quantify upregulation of ZAP-S and ZAP-L isoforms upon infection by analyzing reads mapping to the splice junction. We found that both ZAP-S and ZAP-L are upregulated upon infection, 2.5-fold and 3.57-fold respectively.

Mock: 35 ZAPL fragments and 15 ZAPS fragments

Infected: 125 ZAPL fragments and 38 ZAPS fragments

However, due to the low coverage at the splice junction we consider this data qualitative rather than quantitative. We include these data in the rebuttal for the reviewer's interest.

The ANOVA tests need to be followed by e.g. a Tukey's Posthoc test to actually judge which are the differentially regulated conditions. Applies to several occasions throughout the manuscript.

We apologize for not mentioning that the ANOVA tests were followed by a Brown-Forsythe test to ensure equal variance among the samples. Also, a Dunnett's multiple comparisons test was employed in order to identify the differentially regulated conditions compared to the control constructs using GraphPad Prism version 9.0.0. This is now clarified in the manuscript text.

Figure 2C: How is controlled that the amount of overexpressed protein is correlated to the effect size? The authors place such an emphasis on PRF efficiency to select their candidates that this needs to be controlled.

We acknowledge that there could be differences in their expression levels that may change the observed effect. To account for that, in the dual-fluorescence assay, the putative trans-regulators of frameshifting are expressed as ECFP fusion proteins. Thus, we can apply a gating technique to analyze the frameshifting ratio in cells that are positive for ECFP, and hence express the putative trans-acting regulators (ms lines 150-154). This gating strategy is illustrated in Supplementary Fig. 2A. We now also present the mean ECFP values for a selection of proteins interrogated in this study (Rebuttal Fig. 5). Based on the mean ECFP values expression levels of both ZAP isoforms are actually lower than what we observe for most other candidates. Thus, the data we show in the manuscript likely underestimates the effect of ZAP-S. Nevertheless, we can clearly show expression levels do not differ significantly between ZAP-S and ZAP-L, which in turn also confirms that the difference in effect of ZAP-L and ZAP-S is not due to their expression levels.

Rebuttal Fig. 5 – Mean ECFP intensity for various proteins interrogated in the dual-fluorescence assay as described in Fig. 2 of the manuscript.

Reading through the manuscript, it is not clear to me why ZAP-S should have a stronger effect than ZAP-L, but one explanation might be actual expression levels in this assay here. It would have been a much stronger study if ZAP-L would have been taken along and for a publication at this level, I would like to see the data also for ZAP-L. For me, it looks like a little overselling of the short ZAP isoform, while there is no proof that the long ZAP isoform should not work either. The authors should acknowledge and discuss this better in the manuscript. Especially the comment that “ZAP-S is specific” I don’t see justified with the presented data.

We have now analyzed the mean ECFP intensity of the ZAP-L and ZAP-S fusion proteins (see Rebuttal Fig.5) and demonstrate that the expression levels do not differ significantly between the two isoforms. Originally, we selected the shorter isoform of ZAP for further investigation based on the cellular reporter assays, where a stronger decrease in relative frameshift levels were observed with ZAP-S compared to ZAP-L (see Rebuttal Fig. 6). While we do not yet understand the reasons for this difference, it is possible that the two isoforms have distinct functions, similar to the broad-spectrum frameshift suppressor- Shiftless (SFL), where only one of the two isoforms affects HIV frameshifting (Wang et al., 2019, DOI:/10.1016/j.cell.2018.12.030). Previously, it has also been reported that ZAP-S and ZAP-L have different subcellular localization. While ZAP-S is mainly localized in the cytoplasm, ZAP-L can be prenylated, triggering its membrane-associated. This differential localization was proposed to have a direct influence on the roles these proteins play during infection (Charron et al., 2013, DOI:10.1073/pnas.1302564110; Vyas et al., 2013 DOI:10.1038/ncomms3240; Schwerk et al., 2019, DOI:10.1038/s41590-019-0527-6). Despite our efforts to quantify endogenous levels of both isoforms during infection using available RNA-seq datasets (see point #1 above), we were unable to obtain reliable results due to low coverage of the splice junction.

Rebuttal Fig. 6 – Effect of ZAP-L on FE of SARS-CoV-2, HKU1, OC43 as well as mutants of the SARS-CoV-2 PRF site (compare to ms Fig. 2E).

I also would remind the authors to not overinterpret that only CoV and ZAP worked in their assay. While they tested a few structures, they can't completely rule out that there are more. It is a kind of lame argument on my side, but I don't think that the human ZAP-S isoform just evolved to recognize CoV and CoV2.

Reviewer #1 raised a similar point (pt. 4). As suggested by this reviewer, we tested additional human coronaviruses (namely HKU1 and OC43), but we did not observe a statistically significant reduction in frameshift efficiency (Fig. 2E). However, we agree that ZAP is very unlikely to have evolved to specifically recognize the frameshift motifs of SARS-CoV-1 and -2, given that these are very new human pathogens. We have therefore toned down the manuscript text accordingly.

For Figure 3, the authors should also include data from IGF2BP1 as they seem to suggest that it has a functional relevance and not just IGF2BP3. Why weren't they actually tested in Figure 2C, given they are very similar, but show different effects?

First, we would like to clarify that we had not intended to suggest that IGF2BP1 (IMP1) or IGF2BP3 (IMP3) have functional relevance. We initially included IMP3 as a 'negative' control, because the protein is only very marginally enriched in our interactome screen. Following the suggestions of the reviewer, we have now included both IMP3 and IMP1 in our dual fluorescence reporter assay (Fig. 2C) and show that neither protein is functionally relevant for frameshifting. By comparing various published RNA-seq datasets, we can now also illustrate that IMP1 RNA levels are not upregulated upon infection, while IMP3 levels are increased (Supplemental Fig. 1E).

We also used IMP3 in our *in vitro* optical tweezer analysis, but only as a control to corroborate the specificity of this assay.

Figure 4: For this assay (E and F) IGF2B3 should be used as a control to rule out just general effects of protein in the lysate. The authors already expressed IGF2BP3, thus it is not a large effort to include a more appropriate control.

IGF2BP3 (IMP3) and the SUMO-tag were already used as controls in the *in vitro* translation assay lysates (old supplementary Fig. 2E, now moved to ms Fig. 3C). Additionally, we have now included the purified IMP3 protein as a negative control in the optical tweezers assay (Fig. 5). Our data indicates that IMP3 binds the SARS-CoV-2 frameshifting element with low affinity in MST (ms text line 311-313, Supplementary Fig. 3G) and has no significant effect on frameshifting and the folding of the frameshifting element (Fig. 2C and Fig. 5I).

Why not endogenous studies as a ZC3HAV1 antibody (Fig. S2D) was available? This limits in my view the value of the study. There could be overexpression artefacts.

We are not entirely certain what experiment this comment refers to. Co-immunoprecipitation was performed in infected Calu-3 lysate (removed in the updated ms) and in these experiments, we analyzed endogenous ZAP. However, as an interferon-induced gene, basal cellular ZAP-S levels (in absence of viral infection) are insufficient to be reliably detected by western blotting in a polysome gradient. Hence, ZAP-S had to be overexpressed in those experiments. However, we conducted ribosome pelleting in naïve Calu-3 lysates to show the interactions of endogenous ZAP with ribosomes (Fig. 3F). Nonetheless, to re-focus the manuscript on the *in vitro* characterization of ZAP-S function on frameshifting, as requested by reviewer #1, we de-emphasized the interactions of ZAP-S with the ribosome and removed the ZAP co-immunoprecipitation data from the manuscript (ms lines 261-275).

A orthogonal knock-down/knock-out analysis is also not available.

We assume that this comment refers to the SARS-CoV-2 infection experiments. To address this point, we were able to take advantage of the recently published data by the Kim lab, which shows that depletion of ZAP (in their study referred to as ZC3HAV1) leads to an increase in viral titers (Lee et al., 2021, DOI: 10.1016/j.molcel.2021.04.022) (Rebuttal Fig. 5). We now discuss these findings, which are well in line with our observations, and citation is added in the manuscript text (ms text lines 183-184).

Rebuttal Fig. 7 – Depiction of antiviral effect of ZAP upon knockdown as shown in Lee et al., 2021.

The association with ribosomes has only been done biochemically and looks like a case of sticky behavior. Can this be validated with microscopy or alternatively better suited assay? For sure another RNA-binding protein as negative control is missing in Figure 5b and 5c experiments (actin is not enough).

The initial aim of the ribosome association experiments was to monitor whether ZAP-S interacts with the SARS-CoV-2 RNA during active translation. For that purpose, we would argue that polysome profiling is a suitable and well-accepted assay in the translation field. In addition, ribosome pelleting is generally employed to monitor if a specific factor associates with the ribosomes.

We have now included ribosome pelleting in naïve Calu-3 cells in the main figure to show that endogenous ZAP-S also associates with the ribosomes (Fig. 3F). As an additional control, we also tested ribosome binding of a random RNA binding protein SFPQ. As shown in Rebuttal Fig. 8, compared to the input, SFPQ protein is not significantly enriched in the ribosome pellet, while endogenous ZAP is highly enriched.

Based on comments of reviewer #1, who found our initial presentation of the ZAP-ribosome interactions distracting, we restructured of the manuscript and are presenting these data now in Fig. 3.

We also de-emphasized the observation and functional relevance of the ZAP-ribosome interactions (please also see point above). While we agree that it would be interesting to investigate whether this association occurs in the cell, single RNA localization and single ribosome tracking experiments would be challenging and would require substantial optimizations in cell culture. Therefore, we think that such an in-depth analysis is better suited for follow up work, in line with the suggestions of reviewer #1.

Rebuttal Fig. 8 – Ribosome pelleting of ZAP in naïve Calu-3 cells highlighting that not every RNA-binding protein is enriched in the ribosome pellet (ZAP vs. SFPQ).

Overall: I think that the inclusion of RNA-Seq, adding appropriate controls to experiments in Figure 3, Figure 4 and Figure 5 plus the data on ZAP-L is crucial and should be a prerequisite to accepting this study.

We included additional data and controls in the revised manuscript. Briefly, we analyzed available RNA-seq datasets from infected and uninfected cells as well as patient samples (Blanco-Melo et al. 2020, DOI: 10.1016/j.cell.2020.04.026, Sun et al. 2021, DOI: 10.1016/j.cell.2021.02.008) confirming that ZAP is one of the most highly upregulated transcripts upon infection among the proteins we investigated.

Four new RNA variants were added in ensemble and single molecule experiments to further pinpoint the interactions of ZAP-S with the RNA element. Furthermore, we include additional negative controls (IMP3 as an RNA binding protein and the SUMO tag) to our *in vitro* translation assays and observed no effect on frameshifting levels (Fig. 3C, Supplementary Table 4).

We have also carried out structure probing using DMS-MaPseq, which allowed us to monitor potential ZAP-S-mediated alterations in the RNA structure (Supplementary Fig 4. ms text lines 318-325).

These additional control experiments confirmed the interaction of ZAP-S with the pseudoknot and the effects of this interaction on RNA dynamics.

Minor

I would suggest to rename the heading: “Revealing the short isoform of the host antiviral protein ZAP as an ...” to avoid the misleading information that ZAP-S is a full protein name. Abstract: Also in the text, I would add: “reveal that the short isoform of zinc-finger antiviral protein (ZAP-S) is” (afterwards the use of ZAP-S is justified as introduced as an isoform definition here). Abstract: It is unclear to me why it is a “de novo host encoded factor” and not just a “host encoded factor”.

Wording is changed to illustrate the difference between the two isoforms.

In Figure 1B, the SSB writing is not clear enough visible.

Corrected.

The single-molecule argument might be misleading (see e.g. ZAP interacts with translating ribosome, line 2) as not a single molecule ZAP was studied, but a single RNA molecule.

We apologize for the confusing statement regarding the ZAP-ribosome interaction due to word limits for the abstract. Indeed, the ribosome interaction refers to bulk analysis and not to single molecule experiments. We have modified this sentence to better reflect this point.

Figure S4C: The data is not really interpretable and look more similar than the examples in S4A and S4B. Can this be better represented? I would actually recommend to move Fig. S4D or at least the 0 part to the main figure to not have a cherry-picked representation there, but some global analysis from all measurements.

This point is well taken. Following the suggestion of the reviewer, we have changed the representation from single example traces to show 5 representative curves for folding and unfolding for each sample in Fig. 5D- I. We have also calculated the work from the change in free energies for the wild type and frameshifting RNA variants used in the study. This data is plotted in Fig. 5J, K and histograms for change in contour length and work for each RNA variant is also added in Supplementary Fig. 5 and 6.

REVIEWER COMMENTS

Reviewer #1 (Remarks to the Author):

The authors are to be commended for their thoughtful responses to this referees comments. Great paper!

Reviewer #2 (Remarks to the Author):

The authors may add discussion that (i) ZAP-S has multiple binding modes and may bind to folding intermediate structures and thus slow down the folding rate of the native pseudoknot structure, and (ii) ZAP-S may bind to the native pseudoknot structure and inhibit the interactions of the native pseudoknot structure with ribosome at the frameshift site. Further studies are needed in the future to probe the detailed frameshifting inhibition mechanisms.

Reviewer #3 (Remarks to the Author):

The authors addressed most of my concerns and I think the manuscript can in principle be published. However, the mass spectrometry raw data still needs to be deposited in PRIDE before any publication and the reporting summary needs to be filled out for life science study design when working with cell lines and the used antibodies and eukaryotic cell lines.

Point by Point Replies to Individual Comments

Reviewer #1 (Remarks to the Author):

The authors are to be commended for their thoughtful responses to this referees comments. Great paper!

Great, thanks!

Reviewer #2 (Remarks to the Author):

The authors may add discussion that (i) ZAP-S has multiple binding modes and may bind to folding intermediate structures and thus slow down the folding rate of the native pseudoknot structure, and (ii) ZAP-S may bind to the native pseudoknot structure and inhibit the interactions of the native pseudoknot structure with ribosome at the frameshift site. Further studies are needed in the future to probe the detailed frameshifting inhibition mechanisms.

Thank you very much. We have included the suggested points in the discussion.

Reviewer #3 (Remarks to the Author):

The authors addressed most of my concerns and I think the manuscript can in principle be published. However, the mass spectrometry raw data still needs to be deposited in PRIDE before any publication and the reporting summary needs to be filled out for life science study design when working with cell lines and the used antibodies and eukaryotic cell lines.

Thank you. Mass spectrometry proteomics data have been deposited to the ProteomeXchange Consortium via the PRIDE partner repository with the dataset identifier PXD029656.